## Registered report

statistics

operationalization, methodology, construct validity, measurement, meta science, replication crisis, reproducibility, replicability

**Author for correspondence:**
Matthias Haucke
e-mail: matthias.haucke@gmx.de

# When numbers fail: do researchers agree on operationalization of published research?

## Matthias Haucke[1,3], Rink Hoekstra[2] and Don van Ravenzwaaij[3]

[1]Clinical Psychology and Psychotherapy, Department of Education and Psychology, Freie Universität Berlin, Berlin, Germany
[2]Department of Pedagogical and Educational Sciences and [3]Department of Psychometrics, University of Groningen, Groningen, The Netherlands

 MH, 0000-0002-8761-1124; RH, 0000-0002-1588-7527; DvR, 0000-0002-5030-4091

Current discussions on improving the reproducibility of science often revolve around statistical innovations. However, equally important for improving methodological rigour is a valid operationalization of phenomena. Operationalization is the process of translating theoretical constructs into measurable laboratory quantities. Thus, the validity of operationalization is central for the quality of empirical studies. But do differences in the validity of operationalization affect the way scientists evaluate scientific literature? To investigate this, we manipulated the strength of operationalization of three published studies and sent them to researchers via email. In the first task, researchers were presented with a summary of the Method and Result section from one of the studies and were asked to guess the hypothesis that was investigated via a multiple-choice questionnaire. In a second task, researchers were asked to rate the perceived quality of the study. Our results show that (1) researchers are better at inferring the underlying research question from empirical results if the operationalization is more valid, but (2) the different validity is only to some extent reflected in a judgement of the study's quality. These results combined give partial corroboration to the notion that researchers' evaluations of research results are not affected by operationalization validity.

# 1. Introduction

'Numbers are a creation of our mind. Phenomena do not carry tags with numbers on them, nor do they possess 'quantity' as an intrinsic attribute.'

(Roskam, 1989)

A cornerstone of the credibility of a scientific finding is its replicability [1–4]. However, the replicability of scientific findings remains low in a range of disciplines, such as psychology [5] and biomedicine [6]. So far, suggestions to enhance the reproducibility of findings often revolve around the use of statistics, for example the false interpretation of $p$-values or a lack of control for statistical power [2–4]. One important methodological concern related to replicability that has received little attention in this discussion is the validity of the operationalization.

Operationalization is the process of making a theoretical construct concrete or tangible so that it can be studied via empirical observations [7]. Even if one successfully reduces statistical practices that increase false positives, one can still be faced with erroneous conclusions from falsely mapping empirical results onto constructs. That is, although the results may seem convincing, they do not necessarily warrant the conclusions drawn with respect to one's construct or theory. The goal of this paper is not to investigate which studies have a valid operationalization, but to gather empirical evidence on the extent to which researchers consider the validity of operationalization when drawing conclusions about empirical findings. This is an important question as a study can lead to convincing statistical results, while actually not operationalizing the underlying concept well. Therefore, we believe researchers need to be attuned towards invalid operationalizations. The following study focuses on concepts, which are the basic blocks on which theories stand.

First, we will define the terms *concept*, *measurement* and *operationalization*. A concept is represented by words often taken from everyday conversations (e.g. justice, creativity and intelligence). The more abstract a concept becomes, the less likely it is that researchers will agree on appropriate measurement strategies [7]. Thus, the issue of defining concepts and establishing their relationship to observations is especially relevant in the social sciences, which primarily depend on the investigation of abstract concepts, such as creativity or intelligence. Measurement, on the other hand, is the less abstract process of assigning numbers to observable variables, which can be nominal (e.g. sex), ordinal (e.g. level of education) or continuous (e.g. performance on an intelligence test; [7]). Linking those empirical observations to unobservable concepts is called operationalization [8]. Operationalization is a necessary step to study abstract concepts by defining them in terms of empirical observations. The validity of the theoretical conclusions drawn from data depends on the validity of the operationalization.

To understand the gap between measure and construct, it can be helpful to distinguish between the concept-as-intended and the concept-as-determined [9]. The **concept-as-intended** is one's framework or concept to be investigated (e.g. creativity). On the other hand, the **concept-as-determined** is the actual measurement or empirical variable (e.g. the number of colours used in a drawing task), which serves as the operational definition of the former. To provide empirical support for a theory, the concept-as-determined needs to be a valid representation of the concept-as-intended. If others cannot link constructs to measurement procedures, these procedures and their associated conclusions become less valid. In line with this notion, Feigl stated that 'concepts which are to be of value to the factual sciences must be definable by operations which are… intersubjective and repeatable' [10]. That is, the overlap between the concepts-as-intended and the concepts-as-determined influence the validity of the conclusions that researchers draw from observed data to theory.

Similarly, Cronbach & Meehl [11] adopt the term 'nomological network' to designate the system of law-like relationships that hold between theoretical entities (e.g. intelligence) and their observable indicators (e.g. IQ tests). A set of theoretical statements becomes a system due to the *semantic* overlap of shared terms. We would, for example, have a hard time deriving a person's intelligence from their favourite pop song. According to Meehl [12], a sign of proper operationalization is a high interpersonal consensus on how the theoretical terms are linked to observations.

Consensus on the used operationalization is not always desirable. For instance, Thomas Kuhn argues that scientific advancement happens through violation of consensus, i.e. a paradigm shift [13]. Moreover, McLeod [14] argues that concepts do not only help to advance scientific knowledge by exactly representing aspects of the world, but through their open-endedness and epistemic vagueness. In his view, concepts are not representing theoretical ideas frozen in time but are part of a continual development. In such a way an epistemically vague or fuzzy concept can inspire exact reformulations, as well as the construction of experimental techniques for probing and testing it, if it relates to a

representation specifying its structure and causal nature. We argue that, in order to advance from fuzzy to exact concept, individual researchers need to notice problematic operationalizations (i.e. there is no overlap between methodology and intended hypothesis) such that they should affect the conclusion researchers personally draw from study results. Otherwise, compelling empirical results might prevent researchers from detecting that the underlying concepts are still fuzzy, stifling theory development.

Previous discussions of valid operationalization have centred around the lack of a consistent use of theoretically founded measures, and the stimuli representativeness. Brunswik [15] proposed that 'cues' (or stimuli) should be sampled from the participants' typical environment. He stated that the used cues allow for drawing conclusions about the (non-observable) construct to the extent that the used cues correctly represent the population of environmental stimuli. But even when the stimuli are a valid sample from the stimuli population, previous researchers have noted that the employed measures can lack a clear theoretical foundation, thus increasing the resulting flexibility during data analysis (i.e. allowing selective reporting [16]). This lack of operationalization clarity has been shown and criticized in studies using self-reports [17], experimental studies [18] and fMRI studies [19–21]. For example, Elson *et al.* [18] have demonstrated that the Competitive Reaction Time Task, which is a paradigm that measures aggressive behaviour, allows researchers to operationalize the severity of aggressive behaviour via a noise blast's volume, duration or a composite score of both. Elson and colleagues argue that this lack of a theoretically founded measure makes it easier to report those specific outcome variables that happened to be statistically significant, thus increasing the occurrence of false-positive findings in the literature. In sum, previous research has concentrated on the representativeness of stimuli and the analytical flexibility that results from theoretically unfounded measures.

In the following study, we will not investigate the lack of clear operationalization in specific studies, but we will empirically test researchers' interpretation of these operationalizations. Specifically, we will test whether researchers consider the validity of the operationalization when drawing conclusions about the results of a study. We argue that even when stimuli correctly represent the population of environmental stimuli and there is no variability in operationalizing the construct under investigation, there can be a gap between the measure and the construct that was intended to be measured.

Despite the importance of an appropriate operationalization of concepts, we are not aware of previous studies that examine whether researchers explicitly consider operationalization when evaluating research. To fill this gap, we have conducted a preregistered study in which we investigated to what extent researchers consider the validity of operationalization when drawing conclusions about empirical findings. To do so, we developed three fictional scenarios, which are related to existing research in psychology. The study consisted of two parts. In part 1, we presented empirical outcomes from different studies and assessed whether researchers can reverse engineer the construct under investigation from the used operationalization. In part 2, we investigated whether the validity of the operationalization is related to researchers' perception of the study's quality. Together, the two parts allow us to answer the question whether researchers consider the study's operationalization when drawing conclusions about the quality of empirical results. The Stage 1 registered report, unchanged from the point of in-principle acceptance, may be found at https://osf.io/rgzq5/.

We hypothesized that researchers are less capable of deducing an original hypothesis from a less valid operationalization (less valid condition leads to less correct deduction of original hypothesis). However, we assumed that the less valid operationalization does not affect the perceived quality of a study. Thus, we hypothesized that less operationalization validity does not affect the rating of a study's quality.[1]

# 2. Method

## 2.1. Participants

We have sent emails with a link to an online questionnaire, made with the online survey platform Qualtrics, to the corresponding authors of all articles published in 2015, 2016, 2017, 2018, 2019, 2020 and 2021 from the following journals:

1. Journal of Experimental Psychology: General
2. Psychological Science
3. Journal of Abnormal Psychology

---

[1]In sum, we expect that relatively poor operationalizations have the direct effect of researchers being less capable of reverse-engineering what the underlying research question was, but we expect that those same poor operationalizations do *not* have the indirect effect of researchers rating the study of lower quality.

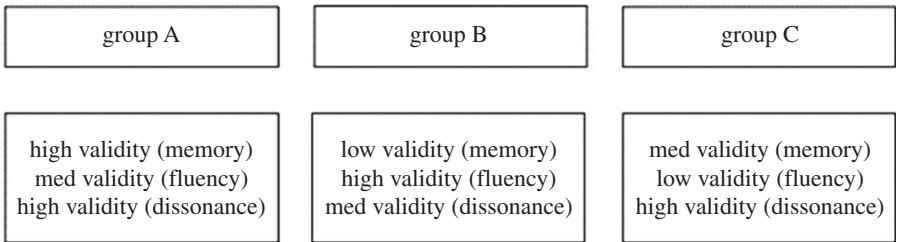

**Figure 1.** Study design. The three groups will first answer a multiple-choice question and subsequently rate each of the three research scenarios. Scenarios presentation within each group will be randomized so that each scenario will appear in varying order.

4. Journal of Consulting and Clinical Psychology
5. Journal of Experimental Social Psychology
6. Journal of Personality and Social Psychology

These journals were chosen to represent a sample of researchers in diverse fields of psychology (experimental, social, neuro and clinical), a sampling strategy previously used by Cramer *et al.* [22]. All duplicates were removed. Participants who did not respond after two weeks received a reminder. Participants were randomly assigned to each of three groups (groups are described in the Research scenarios section below). Text of the invitation email, reminder email, preregistered sampling plan and power calculation can be found at https://osf.io/vsfbh/. We approached 2981 researchers. In total, 325 (10.9%) participants started the online survey, no participant took less than 2 min and 66 participants were excluded due to incomplete responses, resulting in a final sample of 259 participants.

# 3. Materials

## 3.1. Research scenarios

In order to test the influence of operationalization validity on the perceived quality of the study, we adapted three published studies investigating false memory, mental fluency and cognitive dissonance. For each scenario, we manipulate the validity of the operationalization by reducing the strength of the effect, which resulted in three versions that vary in operationalization validity. Each scenario was presented with a different validity condition to each participant (figure 1). Moreover, the presentation of the different scenarios within each group is randomized. There were 85 participants in group A, 90 participants in group B and 84 participants in group C.

### 3.1.1. Scenario 1 (false memory)

In the first scenario, we modified a study by Roediger & McDermott [23]. The study investigated the occurrences of false memory, based on how often participants rated an unseen word to be new or old. Participants were presented with two lists of words they had to memorize. Afterwards, participants had to rate words that were semantically related or unrelated to the words on the memorized list. Then, participants had to rate how likely they thought it was that they had already seen the word (figure 2).

   The results showed that unseen words that were semantically related to the memorized list (i.e. the word '*doctor*' which was not presented in a list of medically related words) increases a false impression of recognition, in comparison to semantically unrelated words. That is, the original study claims that participants falsely remembered '*doctor*', because it's highly similar to the words on the list. Aside from the existing version of the task (labelled 'high validity'), we constructed two other versions of the task by varying the words on the second list to be increasingly semantically similar to the unseen word.'

### 3.1.2. Scenario 2 (perceptual fluency)

In a second scenario, we modified a study by Reber & Schwarz [22]. The study investigated the influence of perceptual fluency on the perceived truthfulness of a statement. Perceptual fluency was defined as the easiness to read a sentence, which was manipulated by using a hard to read or an easy to read colour in a between-subjects design (figure 3).

   The results showed that an easier to read font colour increased the perceptual fluency and thus the truthfulness rating of a sentence (i.e. 'How true is this sentence?'). That is, the original study claims

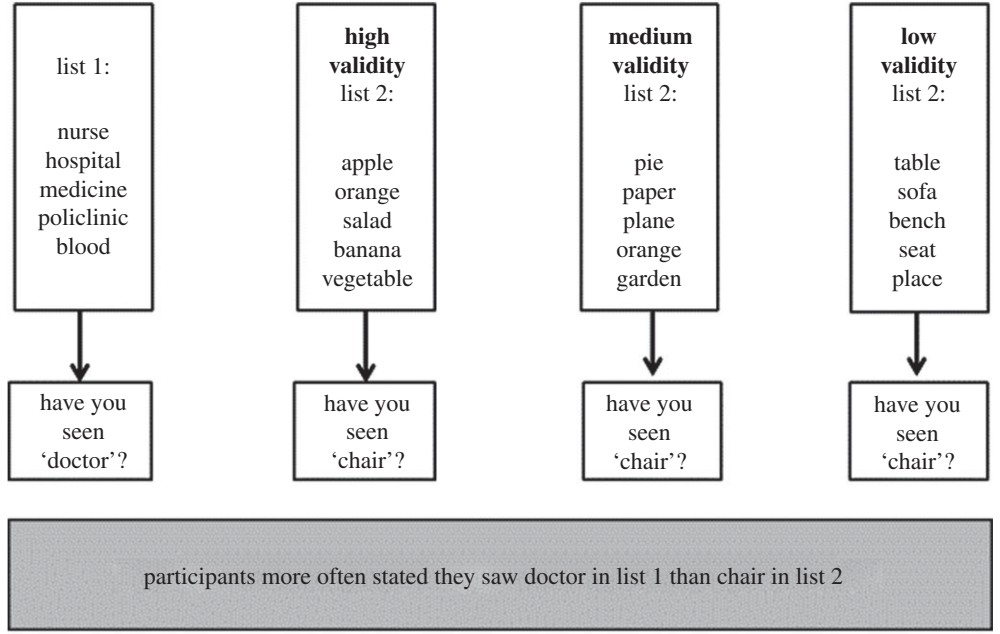

**Figure 2.** Scenario 1. The validity of the operationalization was manipulated by varying the words of list 2 to be semantically unrelated, randomly selected or related to the unseen word.

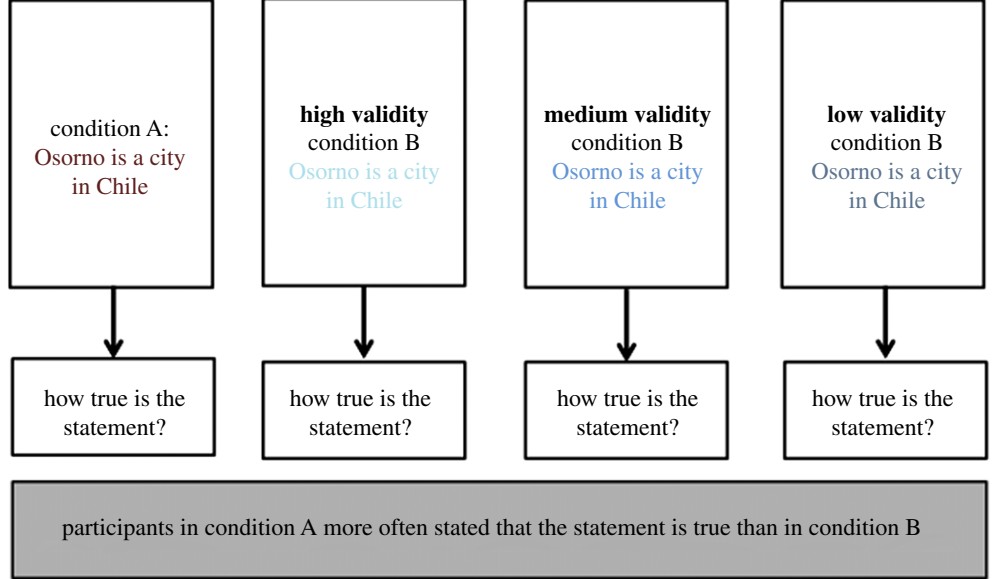

**Figure 3.** Scenario 2. The validity of the operationalization was manipulated by varying the perceptual visibility of the sentence in the second condition.

that participants rated an easy to read font as more truthful, because an easy to read font increases the speed by which textual information could be cognitively processed. Therefore, aside from the existing version of the task (labelled 'high validity'), we constructed two other versions of the task by varying the font colour in the second condition to be increasingly more readable, compared to the first condition. The result is that condition B becomes progressively more similar to read than condition A.

### 3.1.3. Scenario 3 (cognitive dissonance)

In a third scenario, we modified a study by Festinger & Carlsmith [24] on cognitive dissonance. Participants were first asked to do two lengthy and monotonous tasks, which were supposed to create a negative opinion about the tasks. Afterwards, participants were asked by the experimenter to convince another student that the tasks were pleasurable. For doing this, the participant either

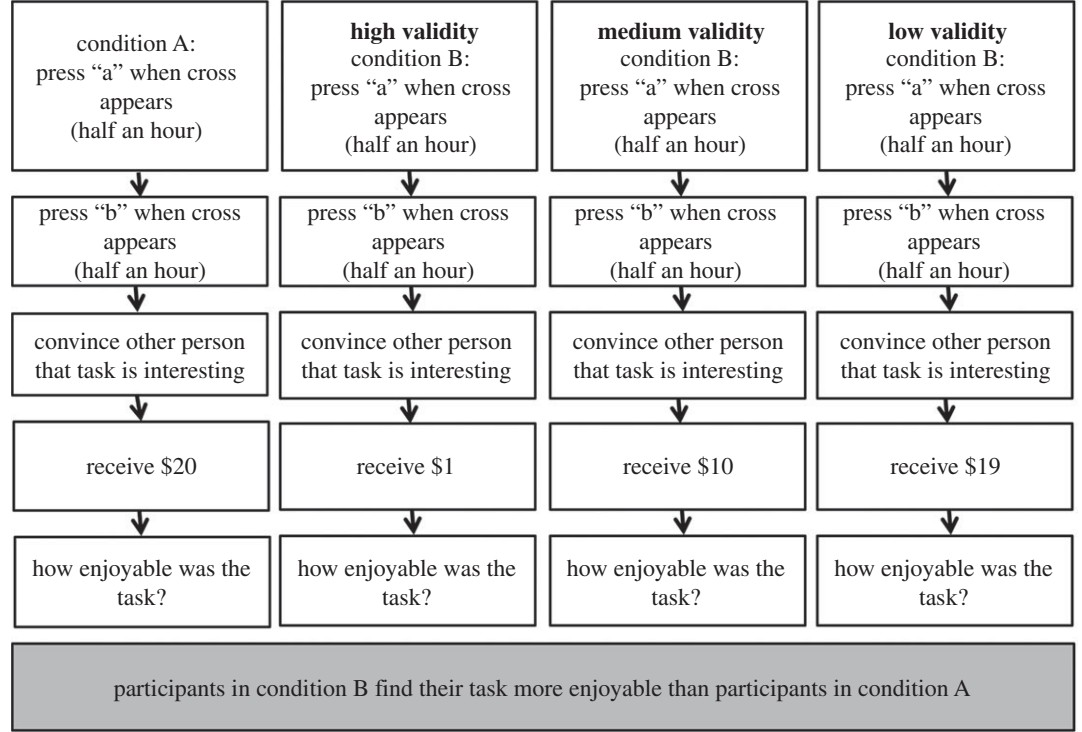

**Figure 4.** Scenario 3. The validity of the operationalization was manipulated by varying the monetary reward in the second condition.

received a high reward (20$) or a small reward (1$). Finally, the participants had to rate how enjoyable they found the monotonous tasks (figure 4).

The results showed that participants who received a high reward rated the tasks as less enjoyable than participants who received a low reward. The authors concluded that participants experienced a dissonance between having to convince a stranger that the tasks were pleasurable, although the tasks were in fact very boring. The cognitive dissonance occurs because participants try to reconcile the fact that they did boring tasks with the fact that it did not even pay well, resulting in a relatively high rating of the perceived enjoyability of the tasks in an attempt to reduce the cognitive dissonance. This dissonance should be smaller in the high reward condition, as the higher pay would be enough reason as and of itself to participate in the tasks without having to find them intrinsically enjoyable.

Aside from the existing version of the task (labelled 'high validity'), we constructed two other versions of the tasks by increasing the monetary reward in the second condition such that it was more similar to the monetary reward in the first condition. The result is that cognitive dissonance should be progressively more similar for condition B compared to condition A for lower validity.

## 3.2. Tasks

### 3.2.1. Task 1: guess the hypothesis

In the first task, we examined whether researchers were more likely to link the methods and results to the research question when the operationalization is more valid via a multiple-choice setting. The presentation consisted of a brief summary of the method and results, as well as their schematic depiction. Each of the three research scenarios was presented as published research and the participants were asked to indicate what they would most likely conclude from the study. The exact questionnaire for each of the research scenarios and validity conditions may be found at https://osf.io/djztx/.

After reading the false memory research scenario, participants were asked to indicate what they thought was a valid conclusion from the presented experiment. They were given four answer choices, of which one was the actual hypothesis that was investigated, whereas the others were incorrect but plausible alternatives: 'Words that are similar to each other can create false memories' (correct), 'Words that are similar to each other can decrease attention to the task', 'Words that are similar to each other can increase creativity' and 'Words that are similar to each other can lead to a wider definition of categories'.

After reading the perceptual fluency scenario, participants were asked to indicate what they thought was a valid conclusion from the presented experiment. They were given four answer choices, of which one was the actual hypothesis that was investigated, whereas the others were incorrect but plausible alternatives: 'Easier to read sentences can increase truth judgements' (correct), 'Red colour can increase truth judgements', 'Blue colour can increase suspicion' and 'Geographical facts can increase truth judgements'.

After reading the cognitive dissonance scenario, participants were asked to indicate what they thought was a valid conclusion from the presented experiment. They were given four answer choices, of which one was the actual hypothesis investigated, whereas the others were incorrect but plausible alternatives: 'Higher mental discrepancy can lead to a change in opinion' (correct), 'Higher monetary rewards can increase negative affect', 'Higher monetary rewards can lead to in-depth cognitive processing of information' and 'Higher cognitive demands can increase negative affect'.

### 3.2.2. Task 2: rate the study quality

After each presented scenario, researchers were asked about the perceived quality of the presented study. Participants answered the following questions: 'How would you judge the quality of the presented study?' and 'How would you judge the strength of support for the theoretical proposition?' on a scale from 1 (very low) to 9 (very high).

## 3.3. Procedure

First, participants were asked to indicate their academic position (BSc, MSc, PhD, Postdoc, Assistant Professor, Associate Professor and Full Professor) and their research background (Social Psychology, Clinical Psychology, etc.). Then, each participant was presented with three research scenarios, described above. For each scenario, participants saw the research methodology and results, but not the research question. Each participant also evaluated the perceived quality of the study. Participation took approximately 15 min. After participants completed the study, we asked them for their familiarity with each presented scenario (1 = not at all, 9 = very high).

# 4. Analysis

We analysed the results of the two tasks with a Bayesian test of proportions [25] for Task 1 and a Bayesian analysis of variance (ANOVA) [26] for Task 2. In the Results section, we reported Bayes factor as well as mean and credible intervals. The Bayes factor ($BF_{10}$) is the relative ratio of the likelihood of the data, given the alternative hypothesis, and the likelihood of the data, given the null hypothesis. For instance, a $BF_{10}$ of 10 indicates that the observed data are 10 times more likely under the alternative hypothesis than under the null hypothesis; a $BF_{10}$ of 1 indicates that the observed data are equally likely under both hypotheses (i.e. the data does not favour one hypothesis over the other) and a $BF_{10}$ of 1/10 indicates that the observed data are 10 times more likely under the null hypothesis than under the alternative hypothesis. In line with the guidelines of Lee & Wagenmakers [27], we interpret a $BF_{10}$ between 1 and 3 as anecdotal evidence, a $BF_{10}$ between 3 and 10 as moderate evidence, 10 and 30 as strong evidence, 30 and 100 as very strong and greater than 100 as extreme evidence in favour of the alternative hypothesis.

## 4.1. Test of three proportions: Task 1

We will modify the model proposed by Kass & Vaidyanathan [25] to account for three proportions:

$$\log\left(\frac{p1}{1-p2}\right) = \beta + \frac{\psi}{2},$$

$$\log\left(\frac{p2}{1-p2}\right) = \beta$$

and

$$\log\left(\frac{p3}{1-p3}\right) = \beta - \frac{\psi}{2}.$$

$$y_1 \sim \text{Binomial } (n_1, p_1)$$
$$y_2 \sim \text{Binomial } (n_2, p_2)$$
$$Y_3 \sim \text{Binomial } (n_3, p_3).$$

In these equations, $p1$ is the proportion of people that correctly pick the true research hypothesis in the highly valid condition, $p2$ is the proportion of people that correctly pick the true research hypothesis in the medium valid condition and $p3$ is the proportion of people that correctly pick the true research hypothesis in the low valid condition. Proportions $p1$ and $p3$ are functions of model parameters $\beta$ and $\psi$. Nuisance parameter $\beta$ corresponds to the grand mean of the log odds, whereas the test-relevant parameter $\psi$ corresponds to the log odds ratio of the two extreme groups. We assigned $\beta$ a standard normal prior and used a zero-centred normal prior with standard deviation $\sigma$ for the log odds ratio $\psi$. The analysis was conducted with Stan [28] and the bridgesampling R package [29]. For ease of interpretation, the results will be shown on the odds ratio scale.

Our analysis primarily focused on testing and quantified the extent to which the data support the null hypothesis: $\psi = 0$ versus the one-sided alternative hypothesis: $\psi > 0$. This one-sided alternative indicates a positive value for $\psi$, resulting in $p1 > p2 > p3$, thus reflecting our hypothesis that a less valid operationalization leads to lower rates of deduction of the intended hypothesis. For the specification of the alternative hypothesis for each presented scenario, we assumed a normal distribution for $\psi$ with mean $\mu = 0$ and standard deviation $\sigma = 0.4$ (i.e. a mildly informative prior; [28]), truncated at zero to take into account that the alternative hypothesis is one-sided, which gives H1: $\psi \sim N (0, 0.4^2)$. We also conducted parameter estimation for the $\psi$ parameter. For this analysis, we used a two-sided prior. The R code for the analysis can be found online at https://osf.io/z4qab/.

## 4.2. Bayesian analysis of variance: Task 2

To test whether the validity of the operationalization is related to the perceived quality of the study, we conducted two univariate three-group between-subjects Bayesian ANOVA with a multivariate generalization of the default Cauchy prior, using the statistical software package JASP [30]. Although we had not planned to do a *post hoc* test, our results made this a valuable additional analysis, so we report these as exploratory. We also conducted an exploratory analysis (Bayesian ANCOVA) including the familiarity of the researcher with the presented scenario.

# 5. Results

Most respondents indicated that their current academic position is Postdoc (26%), followed by Assistant Professor (22%), Associate Professor (20%), Full Professor (17%), PhD (9%) and Other[2] (6%). Most participants indicated that they work in the field of Social Psychology (36%), followed by Clinical Psychology (19%), Cognitive Psychology (14%), Personality Psychology (9%), Other[3] (9%), Experimental Psychology (8%), Methodology and Statistics (2%), Neuroscience (2%), Biological Psychology (0.7%) and Medicine (0.3%).

## 5.1. Preregistered analyses

### 5.1.1. Task 1: guess the hypothesis

The number of correct reverse engineered hypotheses per validity condition can be seen in table 1.

First, we present the results of the hypothesis test. The null hypothesis postulates that there is no difference in correctly deducing the hypothesis between operationalization validity groups H0: $\psi = 0$. The one-sided alternative hypothesis states that the lower the validity of the operationalization, the lower the proportion of correctly deduced hypotheses H1: $\psi > 0$. The Bayes factor indicates overwhelming evidence for H1, with a $BF_{10} = 666665$, which means that the data are over 600 000 times more likely under the alternative hypothesis than under the null hypothesis (figure 5a).

Second, we present the results of the parameter estimation. Of interest is the odds ratio of the high-validity condition against the low-validity condition, defined as (high correct/high incorrect)/(low correct/low incorrect). Figure 5b shows the median of the resulting posterior distribution of odds ratio in the population equals 2.908, with central 95% credible intervals ranging from 2.001 to 4.239, indicating that a population value of 1 (indicating equivalence between the conditions) is very unlikely.

[2]Participants stated that they now work outside of academia.

[3]Participants stated that they work in the fields of developmental/educational psychology, social work, marketing and I/O psychology.

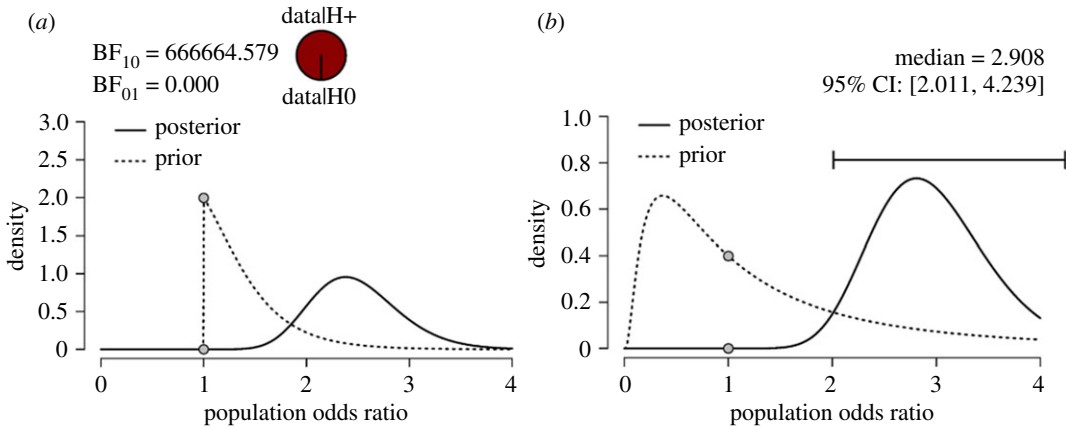

**Figure 5.** Bayesian test of three proportions; (*a*) shows the one-sided procedure for hypothesis testing and (*b*) shows the two-sided procedure for parameter estimation.

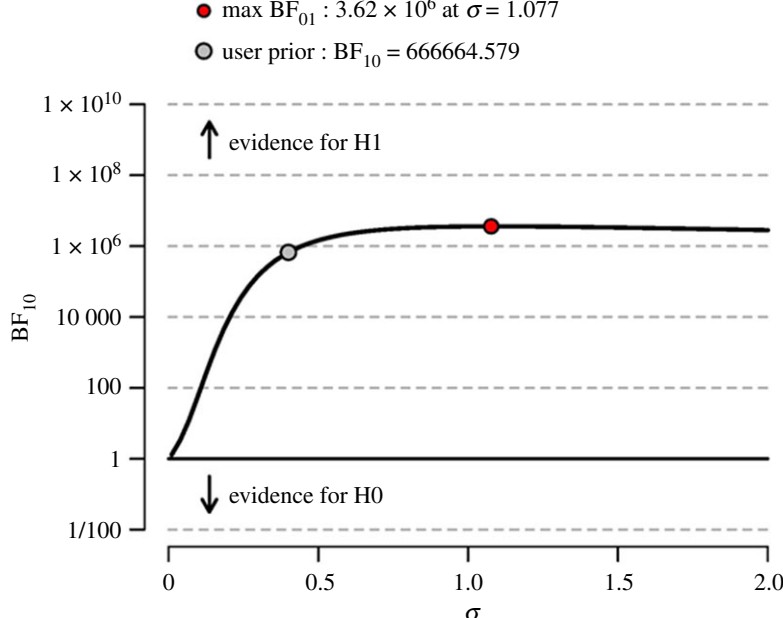

**Figure 6.** The Bayes factor robustness plot. The red dot indicates the prior width setting that results in the maximum $BF_{10}$. The grey dot indicates $BF_{10}$ for the user specified prior ($\sigma = 0.4$).

**Table 1.** Correctly/Incorrectly reverse engineered research hypothesis in Task 1.

| | correct reverse engineering | | |
|---|---|---|---|
| validity condition | no | yes | total |
| high | 56 | 203 | 259 |
| medium | 90 | 169 | 259 |
| low | 117 | 142 | 259 |

To assess the robustness of the Bayes factor to our prior specification, figure 6 shows $BF_{10}$ as a function of the prior width $\sigma$. The Bayes factor appears to be very robust to a wide range of values for $\sigma$ and shows strictly pro-alternative evidence across the entire range that was examined. In sum, the data support our hypothesis that operationalization validity influences the researcher's ability to reverse engineer the correct hypothesis.

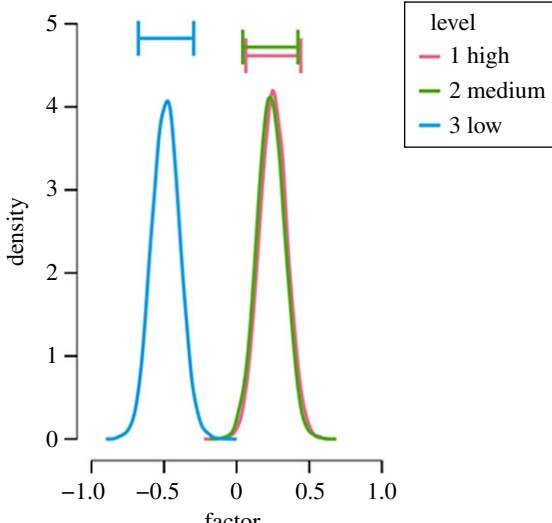

**Figure 7.** The posterior distributions under the alternative hypothesis for each validity condition with perceived quality as an outcome.

**Table 2.** Means (standard deviations in parentheses) of perceived quality and perceived support of proposition per validity condition.

| validity condition | perceived quality (mean(s.d.)) | support of theoretical proposition (mean(s.d.)) |
| --- | --- | --- |
| high | 4.873 (1.866) | 4.293 (2.034) |
| medium | 4.853 (1.949) | 4.305 (1.944) |
| low | 4.108 (1.883) | 3.525 (1.975) |
| total | 4.611 (1.931) | 4.041 (2.016) |

### 5.1.2. Task 2: rate the study quality

A visual check indicates that perceived quality and perceived support of the theoretical proposition are normally distributed and that the assumption of homogeneity of variance holds. The means and standard deviation of both outcomes can be seen in table 2.

First, we present the results of the hypothesis test with the perceived quality of the study as the outcome. The null hypothesis postulates that there is no difference in perceived quality between operationalization validity groups H0: $\delta = 0$. The alternative hypothesis states that the validity of the operationalization is related to the perceived quality H1: $\delta \neq 0$. Contrary to our *a-priori* expectations, we obtained a $BF_{10} = 6154$ which means that the data are over 6000 times more likely under the alternative hypothesis than under the null hypothesis. The posterior distributions under the alternative hypothesis for each validity condition with perceived quality as an outcome are shown in figure 7, reflecting the pattern of the descriptive in table 2 that perceived quality in the low-validity condition differed substantially from the other two conditions.

Second, we present the results of the hypothesis test with perceived support for the theoretical proposition as the outcome. The null hypothesis postulates that there is no difference in perceived support for the theoretical proposition between operationalization validity groups H0: $\delta = 0$. The alternative hypothesis states that the validity of the operationalization is related to the support for the theoretical proposition H1: $\delta \neq 0$. Contrary to our a-priori expectations, we obtained a $BF_{10} = 3861$ which means that the data are over 3000 times more likely under the alternative hypothesis than under the null hypothesis. The posterior distributions under the alternative hypothesis of the average perceived support for the theoretical proposition in the population are shown in figure 8, reflecting the pattern of the descriptive in table 2 that perceived support for the theoretical proposition in the low-validity condition differed substantially from the other two conditions.

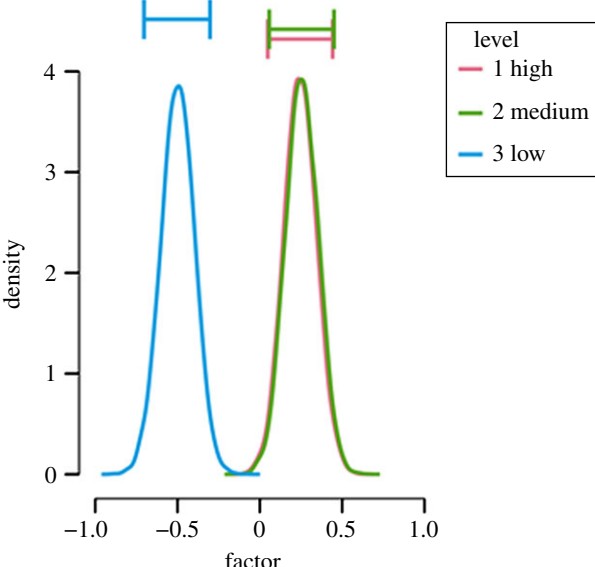

**Figure 8.** The posterior distributions under the alternative hypothesis for each validity condition with perceived support for the theoretical proposition as an outcome.

## 5.2. Interim conclusion

Overall, we found partial support for our *a-priori* expectations. Our data are in line with our first hypothesis: we found that when we reduced the validity of the operationalization, researchers are less able to reverse engineer what the underlying hypothesis was. Contrary to our expectations, we found that researchers also rate the study of lower quality when the validity of the operationalization is reduced. Our original expectation was that compelling results can lead to the perception of a high-quality study even if the mapping of the experiment to the underlying research question is poor.

After observing the data, we do notice an interesting pattern in the results. For Task 1 'Guess the Hypothesis', an inspection of the descriptive suggests results are different for each of the three conditions. For Task 2 'Rate the Study Quality', however, there seems to be a qualitative divide between the low-validity condition on the one hand and the other two conditions on the other hand. It is possible that the manipulation might have been too strong and the unexpected result for our second task might have been driven entirely by the low-validity condition. To test this *post hoc* explanation, we conducted a set of exploratory analyses, in which we essentially repeat our analysis strategy for both tasks, but now focusing exclusively on the comparison of the medium and high-validity conditions.

## 5.3. Exploratory analyses

For the first task, we test whether the proportion of correctly deduced hypotheses differs between the medium- and high-valid conditions. The two-condition analogue to our preregistered analysis is the Bayesian $2 \times 2$ chi-square test, which we conducted using the R package 'BayesFactor' [31]. The null hypothesis states that the medium- and high-validity conditions do not differ in deduced hypothesis H0: $\delta = 0$. The alternative hypothesis states that the validity of the operationalization is related to the proportion of correctly deduced hypotheses H1: $\delta \neq 0$. Under the null hypothesis, the prior for both proportions is a single uniform distribution. Under the alternative hypothesis, each proportion separately has a uniform prior [32]. The Bayes factor indicates strong evidence for the alternative hypothesis, with a $BF_{10} = 25$ which means that the data are approximately 25 times more likely under the alternative hypothesis than under the null hypothesis.

For the second task, the two-condition analogue to our preregistered analysis is the Bayesian *t*-test, which we conducted using the R package 'BayesFactor' [31]. The first *t*-test compared the perceived quality of the study between the medium- and high-validity conditions. The null hypothesis postulates that there is no difference in perceived quality perception between the two operationalization validity groups H0: $\delta = 0$. The two-sided alternative hypothesis states that the validity of the operationalization is related to the perception of quality H1: $\delta \neq 0$. We used a default Cauchy prior distribution with $r = 1/\sqrt{2}$ for effect size parameter $\delta$. In line with our *post hoc*

expectation, we found a $BF_{10} = 0.10$ ($BF_{01} = 1/BF_{10} = 10.2$), which means that the data are approximately 10 times more likely under the null hypothesis than under the alternative hypothesis.

The second $t$-test compared perceived support of the theoretical proposition between the medium- and high-validity conditions. The null hypothesis postulates that there is no difference in perceived support of the theoretical propositions between the two operationalization validity groups H0: $\delta = 0$. The two-sided alternative hypothesis states that the validity of the operationalization is related to the perception of support for the theoretical proposition H1: $\delta \neq 0$. We used a default Cauchy prior distribution with $r = 1/\sqrt{2}$ for effect size parameter $\delta$. In line with our *post hoc* expectation, we found a $BF_{10} = 0.10$ ($BF_{01} = 1/BF_{10} = 10.2$), which means that the data are approximately 10 times more likely under the *null* hypothesis than under the alternative hypothesis.

Finally, using JASP, we conducted two Bayesian ANCOVA with the covariate 'familiarity of with presented scenario' and perceived quality and well as support for the theoretical proposition as outcomes. We report the inclusion Bayes factor (BFincl), which quantifies the change from prior inclusion odds to posterior inclusion odds and can be interpreted as the evidence in the data for including a predictor (i.e. In this case familiarity with the presented scenario) [33]. With perceived quality as an outcome, for Validity Condition, we found BFincl = 332.3 and for Familiarity BFincl = $1.8 \times 10^{26}$. With perceived support for the theoretical proposition as an outcome, for Validity Condition, we found BFincl = 499 and for Familiarity BFincl = $4.6 \times 10^{16}$. Therefore, we can conclude that operationalization validity condition as well as familiarity influenced the perceived quality of the study.

# 6. Discussion

Many findings from the psychological literature are not replicable [5,34], indicating a structural and methodological problem. Most recommendations to address this pertain to publication methods, data collection or data analysis [35–38]. Yet, even the most advanced statistical inferences become redundant if operationalization, the process of translating theoretical constructs into measurable laboratory quantities, fails. Therefore, researchers need to be attuned towards invalid operationalization. In this study, we investigated to which extent researchers consider the validity of operationalization when drawing conclusions about empirical findings.

A sign of proper operationalization is a high interpersonal consensus on how the theoretical terms are linked to observations [12], in line with the preregistered hypothesis we found that researchers are better at inferring the underlying research hypothesis from empirical results in more valid operationalization scenarios. Therefore, we conclude that we have successfully manipulated the validity of the operationalization. We found mixed evidence for the preregistered hypothesis stating that the validity of the operationalization affects the perceived quality of the study. An exploratory analysis shows that researchers were less capable to deduce what the tested hypotheses were in the medium- and high-validity conditions, yet this did not influence either their perception of the study's quality nor the perceived support for the study's theoretical proposition. Thus, we found some support for the notion that the validity of the operationalization does not affect researchers' evaluation of an empirical result.

An extreme change of operationalization validity does impact the researchers' evaluation of empirical results. We found the effect of operationalization validity on the judgement of empirical results only between the medium and high operationalization validity condition, not when we included the low-validity condition. In the low-validity operationalization condition, we reduced the experiment to absurdity (i.e. using the same word category in list A and B in the false memory scenario, using the same colour strength in the perceptual fluency scenario, or giving almost the exact same amount of money in the cognitive dissonance scenario). In the medium validity operationalization condition, on the other hand, the inherent logic of the experiment was preserved (i.e. using different word categories in list A and B in the false memory scenario, using different colour strength in the perceptual fluency scenario, giving substantially different amounts of money in the cognitive dissonance scenario). Thus, we think it is very likely that our manipulation in the low-validity condition was too strong, turning the research scenario into absurdity.

A reason for the lack of researchers' attention to operationalization could be the shared system of beliefs about how psychology works as a science, which influences the dominant methodological practices and the content of methodological education. Psychology is strongly embedded in the experimental tradition, going back to Wilhelm Wundt, which emphasizes empiricism [39]. One way of empiricism to gather objective knowledge about phenomena is the translation of observation into numbers (i.e. making a psychological attribute quantifiable). In line with this, Fechner [40], a pioneer

of experimental psychology, stated '*As an exact science psychophysics, like physics, must rest on experience and the mathematical connection of those empirical facts* (Fechner, 1860, p. xxvii).'

However, the link between concept-as-determined (i.e. actual measurement) and concept-as-intended is non-numerical and subjective. This problem is especially relevant to a range of psychological studies, in which, most concepts-as-intended are highly abstract (e.g. intelligence, empathy, depression). The problem of imprecise concepts-as-intended is further exacerbated by the fact that a majority of published articles do not report the validity of their measured construct, but instead focus on psychometric properties that can be numerically quantified (e.g. reliability measures such as Cronbach's $\alpha$) [41,42]. Moreover, scientists might be hesitant to question established psychological constructs, as constructs are often embedded in 'generative entrenchment', meaning that once a concept has been established (e.g. social anxiety disorder) many other concepts (e.g. fear conditioning), theories (e.g. reinforcement learning) or practices (e.g. cognitive behaviour therapy) depend on it [43]. Finally, there might be a lack of attention on operationalization during the education of psychological researchers. A review of graduate training in psychology has shown that few departments offered a full course on measurement, such as classical test theory (20–24% depending on the topic) [44].

Already in 1967 Paul Meehl stated that there is little theoretical progress in psychological science, with theories tending to come and go without ever being decisively refuted or accepted [12,44]. Michell [45] further argued that the implicit claim that a psychological attribute (e.g. a personality trait, cognitive ability or mental disorder) is quantitative and relates to other attributes quantitatively, needs to be treated as a falsifiable theory. For example, if a researcher tries to predict a person's creativity via their educational background and therefore measure units of creativity, they are accepting the hypothesis that creativity has a quantitative structure (e.g. a creativity unit can be five times another creativity unit); an assumption that may well be false. Therefore, a numerical assignment procedure alone cannot produce scientific measurement. The meaning of a scientific concept and is set by the operations (i.e. the measurement procedures) that were used to identify them [46]. If problematic operationalizations go unnoticed, psychology as a science might get stuck in a process of producing statistically significant results that barely link to their intended construct.

Many voices have been raised calling for theories to be more formal and precise, strengthening the link between theory, construct and hypothesis through mathematical formulation [47–50] and improving publication methods [37]. Yet, mathematical models are simply abstractions of scientific problems, and thus they can aid scientific inference only to the extent that the abstraction is appropriate to the theory and to the concept-as-intended [51,52]. Using more and more precise theoretical formulations can therefore only solve part of the problem and, in the worst case, let us falsely believe that we can somehow 'overcome' the inherent abstract nature of the constructs under investigation. Moreover, improving publication methods does not compensate for weak theory building or low operationalization validity [53]. For instance, why should we judge an operationalization as more valid only because it was preregistered? We believe the real solution is for researchers, editors and reviewers to become more attuned to problematic operationalization.

# 7. Limitations and future studies

In the 'Guess the Hypothesis' task, we used multiple-choice items which impose a context in which the hypotheses are reverse engineered. Thus, we cannot exclude the possibility that other multiple-choice items would have interacted differently with the manipulation of operationalization validity and thus would have changed researchers' capacity to reverse engineer the hypothesis. However, given that our interest was in a relative effect, a difference in proportion correct between different validity conditions, and not in an absolute effect, we do not believe this affects our conclusions.

Related to the previous limitation, researchers who thought that none of the options were valid were not given an option to indicate this. A way around this would have been to include open answer formats. We were concerned that this would make the analysis more subjective, as it would have required us to make judgement calls on whether the descriptions of participants did or did not match the intended hypothesis. However, with multiple raters this may well have been a preferable set-up.

Our study results are limited by the three chosen research scenarios, and it could be that the observed effect is limited to these scenarios. Although we found the reported effects consistently for each of the three research scenarios, future research on this topic could use a wider range of experimental empirical scenarios to present to participants. Moreover, we found that familiarity impacts the perceived quality of a study, thus a fruitful avenue might be to develop completely unknown

scenarios. In addition, future studies could include a wider range of operationalization validity levels than the ones used in our study. Finally, our findings might not be restricted to psychology but apply to any scientific claim that is made using statistical procedures. Therefore, it would be interesting for future studies to investigate whether a similar 'blind eye' to operationalization exists in other disciplines (e.g. economics, medicine).

# 8. Conclusion

Progressing psychological science does not only require advancing statistical inference, but also the successful translation of theoretical constructs into measurable laboratory quantities (i.e. operationalization). Therefore, researchers need to be attuned to empirical studies that have an invalid operationalization. In this study, we found that researchers are better at inferring the underlying research question from empirical results, if the operationalization is more valid. Moreover, only an extreme change of operationalization validity lowers the perceived quality of a presented study. Thus, even if the operationalization of an empirical finding is lowered, it may not affect researchers' perception of the study's quality. Our study indicates that most researchers are not considering the validity of operationalization when evaluating scientific findings.

Ethics. This study was approved by the ethics commission of the University of Groningen (Code 17212-O).
Data accessibility. The article's data, materials and statistical analysis are available on the open science framework at https://osf.io/vsfbh/.
Authors' contributions. M.H. drafted the manuscript, prepared the material, designed the study, gathered the data and analysed the data; R.H. drafted the manuscript, prepared the material and participated in designing the study; D.v.R. drafted the manuscript, prepared the material, designed the study and analysed the data.
Competing interests. The authors of this study have no competing interests.
Funding. D.v.R. was supported by a Dutch scientific organization VIDI fellowship grant (grant no. 016.Vidi.188.001).

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
