## [Peer Review File · Royal Society Open Science]

Review History

RSOS-191354.R0 (Original submission)

Review form: Reviewer 1

Is the language acceptable?

Yes

Do you have any ethical concerns with this paper?

No

Have you any concerns about statistical analyses in this paper?

Yes

Recommendation?

Major revision

Comments to the Author(s)

See attached (Appendix A).

Review form: Reviewer 2

Do you have any ethical concerns with this paper?

No

Recommendation?

Accept in principle

Comments to the Author(s)

Scientific Validity of Question

Studying operationalisation is an important next step for metascience, and understanding replicability. There are lots of interesting questions in this space, such as how often poor operationalisation is at fault for failures to replicate, and how the quality of operationalisation differentially affects direct and conceptual replication success/failure. The current study focuses on whether researchers pay attention to operationalisation. In particular, whether researchers a) more often correctly identify intended hypotheses from more valid operationalisations, and b) whether they rate the study design quality higher when the operationalisation is valid.

Whilst these would not be my choice of question in this space, they seem interesting enough. However, I do think the rationale behind them could be made more explicit. How will this knowledge contribute to our understanding of good versus poor operationalisations, and how they affect the reliability of our research. Why is worthwhile to know the things that are being asked? How do they help us understand the problem of poor operationalisation and its connection to low replicability?

Background discussion might be strengthened by drawing on more recent literature on the role of operationalising constructs (and concepts more broadly). Russell Poldrack and Tal Yarkoni provide a good example of this in the context of neuroimaging experiments in their 2016 paper (see p.590).

p.4 This is a useful distinction, although there must be discussions and extensions of this approach within the relevant literature since the 1970s? There is a related literature on research around 'concepts as used' (rather than just as linguistic or mental representation) - for example see the edited volume *Scientific Concepts and Investigative Practice* (Feest & Steinle, 2012).

The idea that there is a law-like relationship between theoretical entities and their observable indicators is at odds with much contemporary philosophy of science literature. One difficulty is that consensus is not always required for successful operationalisation of a concept (see studies of the uses of the concept of the gene, for example). Of course, consensus seems the only way to set up the current experiment.

Logic, rationale and plausibility of the proposed hypotheses

I have already noted that the rationale for the choice of research questions should be more explicitly articulated. At a high level, both hypotheses seem highly plausible. My concern is, ironically, with how they are operationalised.

Soundness and feasibility of methodology and analysis pipeline (including power analysis)

The manipulation of the operationalisation seems fine in each scenario but I still don't quite understand why there are 4 response options. One is obviously the correct one, but how exactly do the other 3 vary? They are described as being 'plausible alternatives' but do they differ in their plausibility in any ways? Are some better fits with the poorer operationalisations? And why 4 (or rather 3, excluding the correct one)?

I'm also concerned about a ceiling effect of sorts, with the first task, i.e., that the intended hypotheses will be so easy to infer even when it is plainly clear that the operationalisation is poor. In other words, I'm afraid that the first measure won't discriminate. The quality rating task probably will discriminate, but it seems a rough way to measure understanding of or attention to quality. Giving participants the full set of operationalisations and asking them to rank them might give a fuller picture of their understanding, in a more direct way.

There's one further concern, of a different kind. If the text presented to the participants does not include a definition of the construct the instruments are intended to operationalise (to varying degrees of validity) - in this case, false memory - then it's unclear on whether participants will be using the same concept of false memory when choosing from the options in the multiple choice question. If participants draw on different associations with the concept of false memory (as a type of memory), rather than on false memory as a specified construct (for a given cluster of covarying behaviours reliably attributed to the set of standard indicators), then the differences in how they answer the multiple choice question may not relate to the operationalisation of the construct so much as the varying uses of a fuzzy (or undefined) concept. I don't know how likely this is, but it does seem like something that could happen.

Whether the clarity and degree of methodological detail would be sufficient to replicate exactly the proposed experimental procedures and analysis pipeline
This seems sufficiently well covered, but having the exact question wording for task 2 would be preferable.

Whether the authors provide a sufficiently clear and detailed description of the methods to prevent undisclosed flexibility in the experimental procedures or analysis pipeline
The OSF links to code were working, and files are definitely there. I have not closely examined or attempted to run the code, but it's presence offers assurance about this question.

Whether the authors have considered sufficient outcome-neutral conditions (e.g. positive controls) for ensuring that the results obtained are able to test the stated hypotheses
I have already mentioned above a concern about Task 1, and whether it is able to sufficiently test the hypothesis. I also mentioned an alternative set up for Task 2. It's difficult to know what else to suggest, without an explicit rationale for why these particular research questions were chosen.

Review form: Reviewer 3

Do you have any ethical concerns with this paper?

Yes

Recommendation?

Accept with minor revision

Comments to the Author(s)

All comments to the authors can be found in the attached pdf (see Appendix B).

Decision letter (RSOS-191354.R0)

31-Oct-2019

Dear Mr Haucke,

The Editors assigned to your Stage 1 Registered Report ("When numbers fail: Do researchers agree on operationalisation of published research?") have now received comments from reviewers. We would like you to revise your paper in accordance with the referee and editors suggestions which can be found below.

Please submit a copy of your revised paper within three weeks (i.e. by the The author due date is unavailable).

When submitting your revised manuscript, you must respond to the comments made by the referees and upload a file "Response to Referees" in "Section 2 - File Upload". Please use this to document how you have responded to the comments, and the adjustments you have made. In order to expedite the processing of the revised manuscript, please be as specific as possible in your response.

Kind regards,
Lianne Parkhouse
Editorial Coordinatror
Royal Society Open Science
openscience@royalsociety.org

on behalf of Professor Chris Chambers (Registered Reports Editor, Royal Society Open Science)
openscience@royalsociety.org

Associate Editor Comments to Author (Professor Chris Chambers):

At the outset, my apologies for the delay in receiving this decision letter. Three reviewers have now appraised the Stage 1 manuscript. The reviews are encouraging but also indicate a range of issues that will need to be addressed to achieve IPA, including the rationale for the hypotheses (including strength of theory), the level of methodological detail, the suitability of the intervention itself for answering the question, and the adequacy of positive controls. All concerns appear to be readily addressable, therefore the authors are invited to submit a thorough revision. Given the delay in handling the Stage 1 manuscript, I will reach a final decision concerning IPA on the basis of the authors' revised manuscript, without returning the submission to in-depth review.

Comments to Author:

Reviewer: 1

Comments to the Author(s)

See attached.

Reviewer: 2

Comments to the Author(s)

Scientific Validity of Question

Studying operationalisation is an important next step for metascience, and understanding replicability. There are lots of interesting questions in this space, such as how often poor operationalisation is at fault for failures to replicate, and how the quality of operationalisation differentially affects direct and conceptual replication success/failure. The current study focuses on whether researchers pay attention to operationalisation. In particular, whether researchers a) more often correctly identify intended hypotheses from more valid operationalisations, and b) whether they rate the study design quality higher when the operationalisation is valid.

Whilst these would not be my choice of question in this space, they seem interesting enough. However, I do think the rationale behind them could be made more explicit. How will this knowledge contribute to our understanding of good versus poor operationalisations, and how they affect the reliability of our research. Why is worthwhile to know the things that are being asked? How do they help us understand the problem of poor operationalisation and its connection to low replicability?

Background discussion might be strengthened by drawing on more recent literature on the role of operationalising constructs (and concepts more broadly). Russell Poldrack and Tal Yarkoni provide a good example of this in the context of neuroimaging experiments in their 2016 paper (see p.590).

p.4 This is a useful distinction, although there must be discussions and extensions of this approach within the relevant literature since the 1970s? There is a related literature on research around 'concepts as used' (rather than just as linguistic or mental representation) - for example see the edited volume *Scientific Concepts and Investigative Practice* (Feest & Steinle, 2012).

The idea that there is a law-like relationship between theoretical entities and their observable indicators is at odds with much contemporary philosophy of science literature. One difficulty is that consensus is not always required for successful operationalisation of a concept (see studies of the uses of the concept of the gene, for example). Of course, consensus seems the only way to set up the current experiment.

Logic, rationale and plausibility of the proposed hypotheses

I have already noted that the rationale for the choice of research questions should be more explicitly articulated. At a high level, both hypotheses seem highly plausible. My concern is, ironically, with how they are operationalised.

Soundness and feasibility of methodology and analysis pipeline (including power analysis)

The manipulation of the operationalisation seems fine in each scenario but I still don't quite understand why there are 4 response options. One is obviously the correct one, but how exactly do the other 3 vary? They are described as being 'plausible alternatives' but do they differ in their plausibility in any ways? Are some better fits with the poorer operationalisations? And why 4 (or rather 3, excluding the correct one)?

I'm also concerned about a ceiling effect of sorts, with the first task, i.e., that the intended hypotheses will be so easy to infer even when it is plainly clear that the operationalisation is poor. In other words, I'm afraid that the first measure won't discriminate. The quality rating task probably will discriminate, but it seems a rough way to measure understanding of or attention to quality. Giving participants the full set of operationalisations and asking them to rank them might give a fuller picture of their understanding, in a more direct way.

There's one further concern, of a different kind. If the text presented to the participants does not include a definition of the construct the instruments are intended to operationalise (to varying degrees of validity) - in this case, false memory - then it's unclear on whether participants will be using the same concept of false memory when choosing from the options in the multiple choice question. If participants draw on different associations with the concept of false memory (as a type of memory), rather than on false memory as a specified construct (for a given cluster of covarying behaviours reliably attributed to the set of standard indicators), then the differences in how they answer the multiple choice question may not relate to the operationalisation of the construct so much as the varying uses of a fuzzy (or undefined) concept. I don't know how likely this is, but it does seem like something that could happen.

Whether the clarity and degree of methodological detail would be sufficient to replicate exactly the proposed experimental procedures and analysis pipeline

This seems sufficiently well covered, but having the exact question wording for task 2 would be preferable.

Whether the authors provide a sufficiently clear and detailed description of the methods to prevent undisclosed flexibility in the experimental procedures or analysis pipeline

The OSF links to code were working, and files are definitely there. I have not closely examined or attempted to run the code, but it's presence offers assurance about this question.

Whether the authors have considered sufficient outcome-neutral conditions (e.g. positive controls) for ensuring that the results obtained are able to test the stated hypotheses

I have already mentioned above a concern about Task 1, and whether it is able to sufficiently test the hypothesis. I also mentioned an alternative set up for Task 2. It's difficult to know what else to suggest, without an explicit rationale for why these particular research questions were chosen.

Reviewer: 3

Comments to the Author(s)

All comments to the authors can be found in the attached pdf (Review-RSOS-1911354_LT).

Author's Response to Decision Letter for (RSOS-191354.R0)

See Appendix C.

Decision letter (RSOS-191354.R1)

28-Feb-2020

Dear Dr Haucke

On behalf of the Editor, I am pleased to inform you that your Manuscript RSOS-191354.R1 entitled "When numbers fail: Do researchers agree on operationalisation of published research?" has been accepted in principle for publication in Royal Society Open Science. The reviewers' and editors' comments are included at the end of this email.

You may now progress to Stage 2 and complete the study as approved. Before commencing data collection we ask that you:

- 1) Update the journal office as to the anticipated completion date of your study.
- 2) Register your approved protocol on the Open Science Framework (e.g using this registration portal for RRs: <https://osf.io/rr>) or other recognised repository, either publicly or privately under embargo until submission of the Stage 2 manuscript. Please note that a time-stamped, independent registration of the protocol is mandatory under journal policy, and manuscripts that do not conform to this requirement cannot be considered at Stage 2. The protocol should be registered unchanged from its current approved state (in clean rather than tracked-changes form), with the time-stamp preceding implementation of the approved study design.

Following completion of your study, we invite you to resubmit your paper for peer review as a Stage 2 Registered Report. Please note that your manuscript can still be rejected for publication at Stage 2 if the Editors consider any of the following conditions to be met:

- The results were unable to test the authors' proposed hypotheses by failing to meet the approved outcome-neutral criteria.
- The authors altered the Introduction, rationale, or hypotheses, as approved in the Stage 1 submission.
- The authors failed to adhere closely to the registered experimental procedures. Please note that any deviations from the approved experimental procedures must be communicated to the editor immediately for approval, and prior to the completion of data collection. Failure to do so can result in revocation of in-principle acceptance and rejection at Stage 2 (see complete guidelines for further information).
- Any post-hoc (unregistered) analyses were either unjustified, insufficiently caveated, or overly dominant in shaping the authors' conclusions.
- The authors' conclusions were not justified given the data obtained.

We encourage you to read the complete guidelines for authors concerning Stage 2 submissions at <https://royalsocietypublishing.org/rsos/registered-reports#ReviewerGuideRegRep>. Please especially note the requirements for data sharing, reporting the URL of the independently registered protocol, and that withdrawing your manuscript will result in publication of a Withdrawn Registration.

Please note that Royal Society Open Science will introduce article processing charges for all new submissions received from 1 January 2018. Registered Reports submitted and accepted after this date will ONLY be subject to a charge if they subsequently progress to and are accepted as Stage 2 Registered Reports. If your manuscript is submitted and accepted for publication after 1 January 2018 (i.e. as a full Stage 2 Registered Report), you will be asked to pay the article processing charge, unless you request a waiver and this is approved by Royal Society Publishing. You can find out more about the charges at <https://royalsocietypublishing.org/rsos/charges>. Should you have any queries, please contact openscience@royalsociety.org.

Once again, thank you for submitting your manuscript to Royal Society Open Science and we look forward to receiving your Stage 2 submission. If you have any questions at all, please do not hesitate to get in touch. We look forward to hearing from you shortly with the anticipated submission date for your stage two manuscript.

Kind regards,
Andrew Dunn

Royal Society Open Science Editorial Office
Royal Society Open Science
openscience@royalsociety.org

on behalf of Professor Chris Chambers (Registered Reports Editor, Royal Society Open Science)
openscience@royalsociety.org

Author's Response to Decision Letter for (RSOS-191354.R1)

See Appendix D.

RSOS-191354.R2 (Revision)

Review form: Reviewer 2

Is the manuscript scientifically sound in its present form?

Yes

Are the interpretations and conclusions justified by the results?

Yes

Is the language acceptable?

Yes

Do you have any ethical concerns with this paper?

No

Have you any concerns about statistical analyses in this paper?

No

Recommendation?

Accept as is

Comments to the Author(s)

The data provide a fair test of the original hypotheses.

The Introduction, rationale and hypotheses are the same as the approved Stage 1 submission. There are a few very minor wording changes, all of which seem well justified and have been clearly indicated by authors.

Authors adhered precisely to the registered experimental procedures. The few minor exceptions are clearly documented and correct simple errors rather than adding any substantive material. They also had to increase their planned invitation sample size due to a lower response rate. (Low response rates for surveys of researchers have been pretty common over the last 12-18 months.)

Exploratory analyses are clearly indicated and do add some value to the paper.

The authors are well calibrated in their conclusions, for the evidence provided. They provide extensive details about limitations.

Review form: Reviewer 3

Is the manuscript scientifically sound in its present form?

Yes

Are the interpretations and conclusions justified by the results?

Yes

Is the language acceptable?

Yes

Do you have any ethical concerns with this paper?

No

Have you any concerns about statistical analyses in this paper?

No

Recommendation?

Accept with minor revision

Comments to the Author(s)

This was a very interesting study to read and I was looking forward to see the results at Stage 2. I only have a few comments that could be addressed in a minor revision, as detailed in the attached document (Appendix E).

Decision letter (RSOS-191354.R2)

Dear Dr Haucke:

On behalf of the Editor, I am pleased to inform you that your Stage 2 Registered Report RSOS-191354.R2 entitled "When numbers fail: Do researchers agree on operationalisation of published research?" has been deemed suitable for publication in Royal Society Open Science subject to minor revision in accordance with the referee suggestions. Please find the referees' comments at the end of this email.

The reviewers and Subject Editor have recommended publication, but also suggest some minor revisions to your manuscript. Therefore, I invite you to respond to the comments and revise your manuscript.

Please also ensure that all the below editorial sections are included where appropriate -- if any section is not applicable to your manuscript, please can we ask you to nevertheless include the

heading, but explicitly state that the heading is inapplicable. An example of these sections is attached with this email.

- Ethics statement

- Data accessibility

If you wish to submit your supporting data or code to Dryad (<http://datadryad.org/>), or modify your current submission to dryad, please use the following link:
[http://datadryad.org/submit?journalID=RSOS&manu=\(Document not available\)](http://datadryad.org/submit?journalID=RSOS&manu=(Document not available))

- Competing interests

- Authors' contributions

- Acknowledgements

- Funding statement

Because the schedule for publication is very tight, it is a condition of publication that you submit the revised version of your manuscript within 7 days (i.e. by the 29-Jul-2021). If you do not think you will be able to meet this date please let me know immediately.

on behalf of Professor Chris Chambers
(Registered Reports Editor, Royal Society Open Science)
openscience@royalsociety.org

Associate Editor Comments to Author (Professor Chris Chambers):

Comments to the Author:

Two of the original Stage 1 reviewers kindly returned to evaluate the Stage 2 manuscript. As you will see, their comments are overall positive, with Reviewer 2 happy with the manuscript as-is, and Reviewer 3 offering a number of very useful suggestions for minor revision (especially concerning clarifications / deviations from the Stage 1 protocol and delineation of exploratory vs confirmatory analyses). Please attend carefully to these comments in a minor revision.

Comments to Author:

Reviewer: 2

Comments to the Author(s)

The data provide a fair test of the original hypotheses.

The Introduction, rationale and hypotheses are the same as the approved Stage 1 submission. There are a few very minor wording changes, all of which seem well justified and have been clearly indicated by authors.

Authors adhered precisely to the registered experimental procedures. The few minor exceptions are clearly documented and correct simple errors rather than adding any substantive material. They also had to increase their planned invitation sample size due to a lower response rate. (Low response rates for surveys of researchers have been pretty common over the last 12-18 months.)

Exploratory analyses are clearly indicated and do add some value to the paper.

The authors are well calibrated in their conclusions, for the evidence provided. They provide extensive details about limitations.

Reviewer: 3

Comments to the Author(s)

This was a very interesting study to read and I was looking forward to see the results at Stage 2. I only have a few comments that could be addressed in a minor revision, as detailed in the attached document (Review-RSOS-191354.R2.docx).

Author's Response to Decision Letter for (RSOS-191354.R2)

See Appendix F.

Decision letter (RSOS-191354.R3)

Dear Matthias,

It is a pleasure to accept your Stage 2 Registered Report entitled "When numbers fail: Do researchers agree on operationalisation of published research?" in its current form for publication in Royal Society Open Science.

on behalf of Professor Chris Chambers (Subject Editor)
openscience@royalsociety.org

Appendix A

Review of:

When numbers fail: Do researchers agree on operationalization of published research?

Manuscript ID: RSOS-191354

TO THE EDITOR:

Dear editor,

Thank you for the opportunity to review this manuscript.

The research question is interesting and timely. However, there are deficiencies in the design and a lack of transparency about relevant details for a registered report. As such, I think that the manuscript is far from ready for in-principle acceptance in its current form, and unless the design is substantially improved, I don't think that it provides a strong test of whether researchers can tell the quality of an operationalization based on study-design alone.

See below for comments. I have signed my review, for accountability and transparency.

Sincerely,

Leo Tiokhin

TO THE EDITOR/authors:

The authors address an interesting research question: can researchers evaluate the strength of an operationalization when evaluating a research study? They propose to do this by adapting the operationalization from 3 published studies in Psychology, purportedly varying its validity, and testing whether researchers from Psychology 1) can infer the underlying research question and 2) judge the veracity of the empirical conclusions differently given different operationalizations.

I have both major and minor concerns with the submission in its current form. Below, I present this comments, along with an evaluation of the extent to which the submission meets RSOS registered report criteria. I hope that my comments will help the authors to strengthen their registered report proposal.

For accountability and transparency, I would like to sign this review.

Leo Tiokhin

MAJOR CONCERNS

The current proposal does not indicate how the researchers determine whether their manipulation of operationalization strength is valid – what evidence is there that this way of varying the operationalization actually makes the operationalization more or less good? Seems to me that you should at least survey other scientists, or have some sort of external criterion by which you measure operationalization validity.

It's not clear to me that what is actually being manipulated is how "good" the operationalization is, as opposed to how "strong" it is. For example, in scenario 2, the the font color of the sentence makes the sentence easier or harder to read. But in all cases,

the sentence is harder to read than an unmanipulated sentence, you're just manipulating the strength of an effect. So, why is it the case that a "stronger" manipulation is a "better" test of a theory? Imagine if you found an effect with just a weak manipulation. Couldn't this provide stronger evidence for a theory than if you found an effect of the same magnitude but with only the large manipulation? The former tells you that you only need to slightly change a parameter to have a dramatic effect.

Relatedly, it's difficult for me to judge whether an operationalization is good or not if I don't have a clear idea of that the theoretical construct is in the first place. For example, if I had a theory about when people should behave altruistically, I have a very precise theoretical definition of altruism (i.e., a behavior that is personally costly but beneficial to others). This then makes it easier for me to see whether an experiment did a good job operationalizing altruism (e.g., a real-world blood donation vs a hypothetical one in the lab), because the latter is further removed from the theory. In your study, what am I using as my rubric to decide the strength of the operationalization? My intuition?

How can you ensure that there is no floor effect to your manipulation? For instance, I had a hard time understanding the experimental designs based on the text description. It would be better to present people with more visual stimuli (e.g., pictures) to complement the text. How can you ensure that the manipulation actually works? There is no pilot study, and no mention of outcome neutral criteria, so this doesn't give me a way of answering these questions, as the proposal currently stands.

Why is the experimental design not counterbalanced (e.g., the memory task is always first)?

Why is the previously-published operationalization always considered the "high-validity" one? Seems to me that a better way to go would be to have an objective criteria of operationalization strength and then choose the best operationalization based on that. Otherwise you're anchored to the initial study, which may have done things in a shit way. Reminds me of the same discussion people have about doing directly replications of studies when the initial study may have been a bad test of a theory to begin with.

DV's. For the multiple choice DV's, why did you choose these answers? How do I know that the answers you chose don't just make it arbitrarily easy to guess the correct one? Additionally, I don't think that the current multiple-choice design is capturing how researchers actually infer the underlying theory from a design – there is an infinite space of possibilities for theories they could come up with. By narrowing it down to these 4, you're making it much, much easier (potentially) to get the right answer. It's not clear to me that this would generalize at all to the types of situations actually faced by scientists when evaluating designs. Finally, all of the answers are just effects (e.g., words that are similar to each other can create false memories). That's an effect, not a theory. But your research question is whether people can infer something about the operationalization-theory link. So how do you justify this?

Why are participants presented with the results of the study, and then asked to evaluate the methods? We know that people change their inferences about the quality of methods based on just results. Seems to me that a potentially better alternative design would be to present methods, Q1, results, Q2.

Task 2 – says that “we will ask participants to judge the quality of the study”. This seems a bit vague to me. Is there evidence from past work regarding what participants think of when they are asked this question? Or that this is the best way to elicit people’s judgements of study quality?

What inferences will you make when the BF is less than 10 in support of either the alternative or the null hypothesis? Needs to be specified.

How specifically will you determine whether or not the MCMC chain converged? What effective sample size are you looking for? What will you do if it doesn’t converge? Do you have any pilot data to “validate” the statistical models on?

On pages 16-17 the authors indicate that for a large effect size, they need an average of 70 participants for group, but need 450 participants per group for a small effect size. Ok. But then the authors plan to stop collecting data after gathering 640 participants or targeting 6000 participants, even if the BF’s are inconclusive. I have several concerns with this. First, the relevant information is not the average # of participants per group to get a BF of at least 10 in the simulations, but the number of participants needed to get a BF of at least 10 in at least X % of simulations. For example, it could be that the average # is 70 for a medium effect size, but to ensure that you sufficient power to detect a medium effect 90% of the time, you need 140 participants. Second, why set an arbitrary upper limit? Given the current information, I have no idea what percentage of the time this study will provide conclusive evidence in favor or against the null hypothesis.

MINOR CONCERNS

Pg 3 line 46. Unclear. Do you mean “without which no”?

In introduction, may be interesting to comment on the extent to which you need to have good, transparent theory in order to see what operationalization makes sense. For example, for all 3 “theories” being evaluated here, they are all actually “effects” (e.g., “when given a list of words similar to a later presented word, people are more likely to falsely remember having seen the later word”. That’s not a theory – it doesn’t explain the deeper causal mechanisms that then result in this effect, or why humans’ minds are designed this way in the first place.

Pg 6 – what will be the text of the email and the reminder? How long will you wait to decide whether participants can be excluded or included? How specifically will you do random assignment? All of these details need to be specified in an RR proposal.

How do you know that the design will take 15 minutes? Have you piloted this?

Pg 9 – you define perceptual fluency but then go on to talk about cognitive fluency. Clarify?

Pg 8 line 31 – Sorry if this is pedantic, but again, you say “based on the theory” but there is no theory. It’s just an effect”. Less fluent = less true. More fluent = more true. Is this really our bar for what a “theory” is?

REGISTERED REPORT EVALUATION CRITERIA

1. The scientific validity of the research question(s).

The research question (i.e., whether researchers can evaluate the quality of operationalizations of a theoretical construct) is a timely and important one.

2. The logic, rationale, and plausibility of the proposed hypotheses.

The rationale is fine. The research question is interesting. The hypotheses are plausible. The operationalizations are questionable, as is the quality of the design to evaluate it (see comments)

3. The soundness and feasibility of the methodology and analysis pipeline (including statistical power analysis where applicable).

Methods are feasible, but as currently described, inadequate to provide confidence that the study will provide conclusive evidence for or against the hypothesis.

4. Whether the clarity and degree of methodological detail would be sufficient to replicate the proposed experimental procedures and analysis pipeline.

No.

5. Whether the authors provide a sufficiently clear and detailed description of the methods to prevent undisclosed flexibility in the experimental procedures or analysis pipeline.

No.

6. Whether the authors have considered sufficient outcome-neutral conditions (e.g. absence of floor or ceiling effects; positive controls; other quality checks) for ensuring that the results obtained are able to test the stated hypotheses.

No.

Appendix B

Review for RSOS-191354

Overall, I agree that this S1 RR should go forward as it will answer important research questions that can further help us tackle the reproducibility crisis in Psychology and related fields. I believe that the S1 protocol is adequate in terms of details in the methods and analyses sections, but should be refined to achieve a higher level of reproducibility and prevent undisclosed flexibility that could prove problematic at S2. For example, some major revisions I have recommended include adding more details to the study procedure, correcting a potential error in the group assignment procedure and being explicit about potential data exclusions such as handling missing data. More importantly, since this is a S1 RR protocol I think it is absolutely necessary to have a concise hypotheses paragraph somewhere in the paper than can be linked with respective statistical tests without leaving room for subjective interpretation by the reader.

Introduction section

Line 53, p. 3. Repeated use of 'often'- in my opinion it would read better if you replace with "social sciences, which primarily depend on the investigation of abstract concepts" and you could give a simple example (one word), "abstract concepts, such as [...]".

Line 35, p. 4. Very optional, but it would be good to remind the reader in this last sentence of the paragraph that this can undermine the reproducibility (or replicability) of research findings.

Line 47, p. 4. Use of "without" with "no"- replace with "with no.." or only "without". Sentence may need rephrasing. Do you mean to say that it would not be possible for these theoretical statements to become a system..., without deductive fertility and derivation(s?) chains to observational statements? This is quite specific language and sentence could be clearer for the reader to follow. In my opinion, an example would be great in this paragraph, but again it's optional - you could extend the creativity example (what is a derivation chain in this context- put in a parenthesis so the word count isn't affected).

Line 28, p. 5. "the relation"- not entirely clear, maybe replace with relationship between [X] and [Y], or this effect.

Line 46, p. 5. "employed measures" to avoid repetition of "used" (optional).

Line 48, p. 5. I would suggest the addition of a citation in the parenthesis where selective reporting is mentioned, just in case the reader is interested to know more about this QRP.

Line 53, p. 5. have "demonstrated".. which is a paradigm "that measures" (or supposedly measures if this is part of the argument), allows (researchers) to operationalise "the" severity of..

Line 15, p. 6. of "such" or "these" operationalisations

Line 26, p. 6. *concept-as-intended* and *concept-as-determined* terms could be mentioned here too (very optional).

Line 34, p. 6. "published" research to increase specificity (e.g., see line 42, p. 11)- you can also cut this into one sentence, "we are going to conduct a preregistered study that will gather empirical evidence...".

Line 47, p. 6. Very interesting and important research question!

Method section

Line 30, p. 7. Refer reader to section where the group assignment is explained (see X). Also, **major point** here is that Groups A and B are introduced but reading through the next subsection (e.g., as stated in Fig 1 caption) there appears to be three groups in the design (A, B and C), so the group assignment procedure may be incorrect?

Line 22, p. 8. Italicise “doctor”

Line 17, p. 10. Dependent variable may be clearer.

Line 25, p. 15. It would be good to mention what you will tell participants about the aims of the study, as according to instructions or information they may be primed to be more skeptical about the validity of the presented methods in the different scenarios (more relevant for task 2).

Line 8, p. 16. BF explanations are the opposite if you refer to BF_{10} - BF of 1/10 refers to support for the null compared to the alternative hypothesis, and similarly for $BF > 10$, there is support for H1 compared to H0.. unless I missed something here.

Line 31, p. 17. Are you planning to do any follow-up analyses after the ANOVA that could be deemed confirmatory? If yes, these should be stated. Accordingly, what will you do if the ANOVA assumptions are not met? I lack the expertise for the proportions test to comment further on that.

Line 21, p. 18. The use of ‘conclusive’ here suggests that if you end at max N (6000), evidence will be interpreted as inconclusive even if BF for H1, for example, is 8 or 9 (or 10 - this could be ≥ 10 if this is not true). If this is not the case, it should be explained in the S1 protocol to avoid interpretation issues later on.

Line 30, p. 18. Remove quotation mark.

Other comments on methods & analyses.

Major issues

- 1. Missing data, outliers.** Although it may not seem fundamental to the S1 protocol, it is important to account for all possibilities when processing and analysing the data. From the methods section I wasn't sure if all responses will be required (i.e., they cannot proceed if they haven't answered a question) for participants. Are you going to treat participants who consistently respond in the same way across all scenarios as outliers? Are you planning to clean the data in any way (e.g., incomplete responses)? Even if missing data or outliers are not handled in any way, this needs to be clearly stated in an S1 RR protocol.
- 2. Study procedure.** From the methods section I was also not entirely confident to reproduce the study-are participants answering a survey online? If yes, how is this administered (are you using specific software)? Are the three scenarios presented in random order across participants or in fixed order (1. Memory, 2. Fluency, 3. Dissonance)? Please add more details if possible about how participants conduct these tasks after they have been reached via email.
- 3. Hypotheses.** I understand that you have specified your research questions and planned analyses, but I am missing a hypotheses subsection or related statements. Even if these are indirectly stated in the methods, it is important for RRs to have a specific paragraph with stated confirmatory hypotheses (a priori), which relate to exclusive statistical tests.

Minor issues

4. **Other.** How do we account for the confound of experience? Authors that are more senior may have knowledge about the observed effects and related theory. I think you may need to ask them directly after completion- were you aware of this task or related theory and research for task 1 (multiple choice - hypotheses). This relates to point 1 (are you going to exclude non-naive participants)?
5. **Consistency.** The protocol mentions that the study will have two parts and that participants will answer a multiple-choice question and subsequently rate the quality of the study. However, on line 17 p. 17 you mention “specification of H1 for both experiments”, which I found confusing - what are the two experiments - this may suggest that some participants will only do part one and others the second part. The participant-related sections (Sampling plan, Participants, Study procedure) are not in close proximity in the paper so I would suggest moving the Sampling plan closer to the Participants paragraph and refer to relevant analyses in the main text (see X for details on Bayesian analyses), but this is entirely optional.
6. **Test of three proportions.** This section was a bit hard to follow but it is great that you provided all the important details for the test in line with RR guidelines. The only issue I found here relates to my third comment on Hypotheses. You have the description shown below, but as a reader I would like to know how this reflects your hypothesis descriptively - e.g., the proportion of participants who pick the correct hypothesis [...]. Not clearly stating your hypothesis in relation to your planned test leaves room for interpretation which does not ensure the highest level of reproducibility. Thankfully, this is easy to fix by adding a small subsection on specific hypotheses.

“Our first analysis will focus on estimation and uses $\sigma = 1$. Our second analysis will focus on testing and quantifies the extent to which the data support the $H_0: \psi = 0$ versus the **one-sided H1** : $\psi > 0$. This one-sided alternative indicates a positive value for ψ , resulting in $p_1 > p_2 > p_3$ ”

Dear Reviewers,

We would like to thank you for your helpful comments. In the following document you can find each of our response to each point that has been raised.

With kind regards,

Matthias Haucke, Rink Hoekstra and Don van Ravenzwaaij

Reviewer 1:

MAJOR CONCERNS

1. The current proposal does not indicate how the researchers determine whether their manipulation of operationalization strength is valid – what evidence is there that this way of varying the operationalization actually makes the operationalization more or less good? Seems to me that you should at least survey other scientists, or have some sort of external criterion by which you measure operationalization validity.

Thank you for this comment. In the revised manuscript, we outlined in more detail why our manipulations make each of the concepts less applicable to the study method and results at hand. The three studies now read as follows (added parts highlighted):

For false memory:

“The results showed that unseen words that were semantically related to the memorized list (i.e., the word ‘*doctor*’ which was not presented in a list of medically related words) increases a false impression of recognition, in comparison to semantically unrelated words. That is, the original study claims that participants falsely remembered ‘*doctor*’ because it’s highly similar to the words on the list. Thus, by increasing the similarity of the other list (until it matches for low validity), we create a false recognition effect that is progressively weaker. Aside from the existing version of the task (labelled ‘high validity’), we constructed two other versions of the task by varying the words on the second list to be increasingly semantically similar to the unseen word (see Figure 2).”-> **p. 9**

Cognitive fluency:

“The results indicated that an easier to read font colour increased the perceptual fluency and thus the truthfulness rating of a sentence (i.e. “How true is this sentence?”). That is, the original study claims that participants rated an easy to read font as more truthful, because an easy to read font increases the speed by which textual information could be cognitively

processed. Therefore, by progressively increasing the visibility of a font colour, we create more readable sentences, for which information will be easier to process, and the proposed cognitive fluency concept will be less applicable. Therefore, aside from the existing version of the task (labelled 'high validity'), we constructed two other versions of the task by varying the font colour in the second condition to be increasingly more readable, compared to the first condition (see Figure 3). The result is that Condition B becomes progressively more similar to read to Condition A."-> p. 10

Cognitive dissonance:

"It was found that participants who received a high reward rated the task as less enjoyable than participants who received a low reward. The authors concluded that participants experienced a dissonance between having to convince a stranger that the task was pleasurable, although the task was in fact very boring. The explanation for this change in opinion was that by changing towards a more favorable opinion the experienced dissonance between the boring task and the small reward became smaller. In the larger reward context, however, the dissonance was smaller (it's fine to be paid a lot for a boring task), thus making it less necessary to adjust the opinion about the task. Accordingly, cognitive dissonance occurs because participants engaged in an action (i.e. convince a stranger of a pleasurable task) that goes against one of his/her beliefs (i.e. "task was boring, and payment was terrible"), this dissonance resulted in an attitude change (i.e. "task was indeed exciting"). We progressively decreased the experienced dissonance by the participants by increasing the monetary reward in the second condition, thus making the concept of cognitive dissonance less applicable to the results at hand."-> p. 11/12

2. It's not clear to me that what is actually being manipulated is how "good" the operationalization is, as opposed to how "strong" it is. For example, in scenario 2, the font color of the sentence makes the sentence easier or harder to read. But in all cases, the sentence is harder to read than an unmanipulated sentence, you're just manipulating the strength of an effect. So, why is it the case that a "stronger" manipulation is a "better" test of a theory? Imagine if you found an effect with just a weak manipulation. Couldn't this provide stronger evidence for a theory than if you found an effect of the same magnitude but with only the large manipulation? The former tells you that you only need to slightly change a parameter to have a dramatic effect.

This is true, we manipulated the strength of the manipulation by reducing the strength of the effect. In our answer to point 1 we explain why we think it lowers the validity of each of the three operationalisations. Although we cannot be sure that the medium validity condition represents a weaker effect than the high validity effect in all cases, based on theoretical grounds the effect should disappear in the low validity group for each of the three experiments. To acknowledge the point of the reviewer, we made this point clearer in the method section by adding the following sentence.

"For each scenario we manipulate the validity of the operationalization by reducing the strength of the effect, which resulted in three versions that vary in operationalisation validity." -> p. 7

3. Relatedly, it's difficult for me to judge whether an operationalization is good or not if I don't have a clear idea of that the theoretical construct is in the first place. For example, if I had a theory about when people should behave altruistically, I have a very precise theoretical definition of altruism (i.e., a behavior that is personally costly but beneficial to others). **This then makes it easier for me to see whether an experiment did a good job operationalizing altruism (e.g., a real-world blood donation vs a hypothetical one in the lab), because the latter is further removed from the theory. In your study, what am I using as my rubric to decide the strength of the operationalization? My intuition?**

We agree with the reviewer that it is easier to evaluate an operationalization if we know about the theoretical construct. However, we fear that when a theoretical construct is provided, researchers often take the proposed operationalization for granted. By removing the theoretical construct, but presenting the operationalization with differing degrees of validity, we examine specifically if researchers are susceptible to differences in operationalizations in terms of their inference as to what is being investigated.

4. How can you ensure that there is no floor effect to your manipulation? For instance, I had a hard time understanding the experimental designs based on the text description. It would be better to present people with more visual stimuli (e.g., pictures) to complement the text.

This is a very useful suggestion. We realize that our study is about quite an abstract topic, and adding figures typically helps to understand complex stories. To increase understanding of the task, we included pictures of the tasks in Figures 5, 6, and 7 and of the different levels of validity in Figures 2, 3, and 4. Thanks to the addition of these figures, we think the design is now easier to grasp for readers.

5. How can you ensure that the manipulation actually works? There is no pilot study, and no mention of outcome neutral criteria, so this doesn't give me a way of answering these questions, as the proposal currently stands.

We manipulated the connection between construct and results based on the three relevant theories. There is no manipulation in the sense that we change researchers' attitudes or perceptions, we are asking researchers' what they infer from a certain operationalization. Whether or not this differs between validity conditions is the very research question we are trying to answer with this study.

6. Why is the experimental design not counterbalanced (e.g., the memory task is always first)?

Because of this point, we reevaluated our design, and we agree with the reviewer that counterbalancing would be preferable. We adapted the design so that the

scenarios, as well as the validity conditions are counterbalanced. Below, the added text is shown explaining the counterbalancing procedure,

“Moreover, presentation of the different scenarios within each group is randomized.”

Figure 1. Study design. The three groups will first answer a multiple-choice question and subsequently rate each of the three research scenarios. Scenarios presentation within each Group will be randomized so that each scenario will appear in varying order.”

-> p.8/9

7. Why is the previously-published operationalization always considered the “high validity” one? Seems to me that a better way to go would be to have an objective criteria of operationalization strength and then choose the best operationalization based on that. Otherwise you’re anchored to the initial study, which may have done things in a shit way. Reminds me of the same discussion people have about doing directly replications of studies when the initial study may have been a bad test of a theory to begin with.

We agree that there is no objective criterion. But all these are published studies that went through a peer-review process, so, at least at first, it does seem a decent starting point to assume that these are more valid operationalizations than the ones we purposely manipulate to be invalid. So, although we agree that relying on original studies is not guaranteed to be a safe bet, we believe the published literature is the most defensible indicator of the strength of an operationalization available.

8. DV’s. For the multiple choice DV’s, why did you choose these answers? How do I know that the answers you chose don’t just make it arbitrarily easy to guess the correct one?

To give a short answer, we simply made the manipulation weaker up to the point that the effect, in our view, becomes negligible. For a more extensive explanation, we refer to the answers to 1 and 2.

9. Additionally, I don’t think that the current multiple-choice design is capturing how researchers actually infer the underlying theory from a design – there is an infinite space of possibilities for theories they could come up with. By narrowing it down to these 4, you’re making it much, much easier (potentially) to get the right answer. It’s not clear to me that this would generalize at all to the types of situations actually faced by scientists

when evaluating designs.

We will be careful in generalizing our results to the entire population of all possible theories. The important comparison in our study, however, is between the different validity conditions, so we will not focus on the differences in proportion correct across different study scenarios, but only on the differences in proportion correct within study scenarios for different validity conditions. The multiple-choice format has an objective scoring criterium, making it much easier to compare different conditions.

10. Finally, all of the answers are just effects (e.g., words that are similar to each other can create false memories). That's an effect, not a theory. But your research question is whether people can infer something about the operationalization theory link. So how do you justify this?

In the previous manuscript we may not have been clear enough about the difference between theory and construct. In our manuscript, we are not looking for theories but constructs (e.g. memory). A construct would be on a lower level than a theory. On several occasions in the manuscript, we changed occurrences of "theory" by "construct" .

11. Why are participants presented with the results of the study, and then asked to evaluate the methods? We know that people change their inferences about the quality of methods based on just results. Seems to me that a potentially better alternative design would be to present methods, Q1, results, Q2.

We understand why the reviewer suggests this: there are definitely advantages of that order. However, we think this design decision would make our intentions obvious to the participants, which would undermine the entire goal of the study. We think a lot of researchers would not appreciate an obvious manipulation, potentially resulting in answering in a different way than they would naturally do.

12. Task 2 – says that “we will ask participants to judge the quality of the study”. This seems a bit vague to me. Is there evidence from past work regarding what participants think of when they are asked this question? Or that this is the best way to elicit people’s judgements of study quality?

We are not aware of previous studies that have asked for researchers' perception of the quality of scientific literature, which unfortunately makes it impossible to rely on previous materials. We explicitly stated the wording of the two questions now:

“After each presented scenario, we will ask the researcher about the perceived quality of the presented study. Participants will answer the following questions: “ How would you judge the quality of the presented study?” and “ How would you judge the strength of support for the theoretical proposition?” on a scale from 1 (very low) to 10 (very high).” ->p. 17

13. What inferences will you make when the BF is less than 10 in support of either the

alternative or the null hypothesis? Needs to be specified.

We now added a guideline with which we will evaluate BF (including lower than 10):

“In line with the guidelines of Lee and Wagenmakers (29), we will interpret a BF between 1-3 as anecdotal evidence, a BF between 3-10 as moderate evidence, 10-30 as strong evidence, 30-100 as very strong and > 100 as extreme evidence in favour for H_1/H_0 .” -> **p. 17/18**

14. How specifically will you determine whether or not the MCMC chain converged? What effective sample size are you looking for? What will you do if it doesn't converge?

For ANOVA the integral only goes over one parameter, so full MCMC is not needed. The Bayes factor package uses a rough heuristic to choose between simple Monte Carlo integration and a custom importance sampling method.

15. Do you have any pilot data to “validate” the statistical models on?

We do not have pilot data (see also point 5.), but ANOVAs are a very common and established statistical model to analyze this kind of data. Perhaps the reviewer means something else?

16. On pages 16-17 the authors indicate that for a large effect size, they need an average of 70 participants for group, but need 450 participants per group for a small effect size. Ok. But then the authors plan to stop collecting data after gathering 640 participants or targeting 6000 participants, even if the BF's are inconclusive. I have several concerns with this. First, the relevant information is not the average # of participants per group to get a BF of at least 10 in the simulations, but the number of participants needed to get a BF of at least 10 in at least X % of simulations. For example, it could be that the average # is 70 for a medium effect size, but to ensure that you sufficient power to detect a medium effect 90% of the time, you need 140 participants. Second, why set an arbitrary upper limit? Given the current information, I have no idea what percentage of the time this study will provide conclusive evidence in favor or against the null hypothesis.

Thank you for this comment. Indeed, in our power analysis we did not use the average # of participants, but the minimum # for 80% of the simulations resulting in BF higher or equal to 10. We changed this in the text:

“Our current power analysis is based on the univariate three group between-subjects Bayesian ANOVA. To estimate how many participants are needed, we conducted a sequential Bayesian design analysis using a Monte Carlo simulation (see R code in Appendix). In this design, we compute the BF for a simulated data set, while increasing sample size in steps of 1, using a uniform prior. Then, we calculate the minimum sample size for getting at least 80% of BF higher or equal to 10, based on all 10,000 simulations. Assuming a H_1 threshold of >10 , and a medium overall effect size of $f = 0.28$, we need a minimum sample size of 70 participants per group. For a small overall effect size of $f = 0.15$,

we need a minimum sample number of 450 participants per group. Finally, assuming a H0 and a threshold of $<1/10$, with an effect size of $f = 0.0$, we need an average sample size of 160 participants per group." -> p. 19/20

MINOR CONCERNS

17. Pg 3 line 46. Unclear. Do you mean "without which no"?

Yes. Thanks for pointing this out. This grammar mistake has been fixed.

"A set of theoretical statements become a system due to their semantic overlap of shared terms, it would not be possible for theoretical statements to become a system, without deductive fertility. Without deductive fertility derivation chains to observational statements would not be possible." -> p. 4

18. In introduction, may be interesting to comment on the extent to which you need to have good, transparent theory in order to see what operationalization makes sense. For example, for all 3 "theories" being evaluated here, they are all actually "effects" (e.g., "when given a list of words similar to a later presented word, people are more likely to falsely remember having seen the later word". That's not a theory – it doesn't explain the deeper causal mechanisms that then result in this effect, or why humans' minds are designed this way in the first place.

We agree that solely focusing on an effect is at most a very rudimentary theory, and only when using a very lenient definition. What we try to focus on are concepts and we have tried to do that more consistently than in the previous version of the manuscript. (See also our response to point 10)

19. Pg 6 – what will be the text of the email and the reminder? How long will you wait to decide whether participants can be excluded or included? How specifically will you do random assignment?

Thank you for this point, we added this information to the Method section.

"These journals were chosen to represent a sample of researchers in diverse fields of psychology (experimental, social, neuro, and clinical), a sampling strategy previously used by Cramer et al. (23). All duplicates will be removed. Participants who do not respond after two weeks will receive a reminder. One third of the participants will be randomly assigned to Group A, one third will be assigned to Group B and the last third will be assigned to group C. We will randomize the email list, so that approximately each group gets assigned the same number of email addresses. The invitation email will state: "Dear (specified name), we are researchers at the university of Groningen (Netherlands) and are interested in investigating how researchers evaluate studies. The following link will advance you to an online experiment, in which you will be presented with three different studies. If you should choose to participate, your task would be to link the presented study to the correct hypothesis and judge the quality of the study. Participation will take approximately 15 minutes. We would appreciate your response to our invitation. With kind regards (name of researchers)." The reminder email will state the same text, starting with: "Dear (specified name), hereby we

would like to send you a reminder for our study investigating your perception of three different studies. Please follow this link ...”.

“Participants who took less than 2 minutes to fill in the study and who did not finish the study will be excluded. We will make every question mandatory for proceeding. We will not exclude participants who are familiar with the presented studies.”

-> p. 7/8

20. All of these details need to be specified in an RR proposal.

How do you know that the design will take 15 minutes? Have you piloted this?

We have piloted the task and it took on average 8 minutes to finish. However, we think that some participants might take a bit longer, thus we think 15 minutes to be a reasonable estimate of the maximum time it will take to fill in.

21. Pg 9 – you define perceptual fluency but then go on to talk about cognitive fluency. Clarify?

Perceptual fluency is claimed to be a form of cognitive fluency. However, to avoid confusion we stick to perceptual fluency and explain the concept in more detail:

“Scenario 2 (perceptual fluency). In a second scenario, we modified a study by Reber and Schwarz (24). The study investigated the influence of perceptual fluency on the perceived truthfulness of a statement. Perceptual fluency was defined as the easiness to read a sentence, which was manipulated by using a hard to read or an easy to read colour in a between-subjects design. The results indicated that an easier to read font colour increased the perceptual fluency and thus the truthfulness rating of a sentence. That is, the original study claims that participants rated an easy to read font as more truthful, because an easy to read font increases the speed by which textual information could be cognitively processed. Therefore, by progressively increasing the visibility of a font colour, we create more readable sentences, whose information will be easier to process, and the proposed cognitive fluency concept will be less applicable. Aside from the existing version of the task (labelled ‘high validity’), we constructed two other versions of the task by varying the font colour in the second condition to be increasingly more readable, compared to the first condition (see Figure 3). The result is that Condition B becomes progressively more similar to read to Condition A.” -> p. 10

22. Pg 8 line 31 – Sorry if this is pedantic, but again, you say “based on the theory” but there is no theory. It’s just an effect”. Less fluent = less true. More fluent = more true. Is this really our bar for what a “theory” is?

This is not pedantic, and the reviewer is completely right: This is indeed not a theory, but an effect. We changed the wording there.

“The results showed that unseen words that were semantically related to the memorized list (i.e., the word doctor which was not presented in a list of medically related words) increases

a false impression of recognition, in comparison to semantically unrelated words. That is, the original study claims that participants falsely remembered 'doctor' because it's highly similar to the words on the list. Thus, by increasing the similarity of the other list (until it matches for low validity), we create an effect that is progressively weaker and thus becomes progressively unrelated to the original concept of false recognition. Therefore, aside from the existing version of the task (labelled 'high validity'), we constructed two other versions of the task by varying the words on the second list to be increasingly semantically similar to the unseen word (see Figure 2).“ -> **p. 9**

Reviewer 2

1. I do think the rationale behind them could be made more explicit. How will this knowledge contribute to our understanding of good versus poor operationalisations, and how they affect their reliability of our research. Why is worthwhile to know the things that are being asked? How do they help us understand the problem of poor operationalisation and its connection to low replicability?

We agree. We adapted this in our text to make the position clearer:

“Thus, the validity of operationalisation is central for the quality of empirical studies. But do differences in the validity of operationalisation affect the way scientists evaluate scientific literature?” -> **p. 1**

“The goal of this paper is not to investigate which studies have a valid operationalisation, but to gather empirical evidence on the extent to which researchers consider the validity of operationalisation when drawing conclusions about empirical findings. This is an important question as a study can lead to convincing statistical results, while actually not operationalising the underlying concept well. Therefore, researchers need to be attuned towards invalid operationalisation.” -> **p.2**

“Firstly, poor operationalisation lowers the success rate of direct replications because we expect a higher amount of findings by chance (i.e. product of random variation or measurement error), when an effect is unfounded by theory (i.e., measurement barely links to theory), than an effect that is theoretically well founded (i.e., measurement links well to theory). Secondly, poor operationalisation lowers the success rate of conceptual replications because it will lead to scientific claims based on measurements that are barely related to a concept and/or theory. Therefore, the observed effect might not be explained by an intended theoretical underpinning. Instead, a different explanation underlies the relation, which disappears when changing one's measurement and experimental procedure.” ->**p.3**

2. Background discussion might be strengthened by drawing on more recent literature on the role of operationalising constructs (and concepts more broadly). Russell Poldrack and Tal Yarkoni provide a good example of this in the context of neuroimaging experiments in their 2016 paper (see p.590).

Thank you for this interesting article We added it as reference 22:

“This lack of operationalisation clarity has been shown and criticized in studies using self-reports (18), experimental studies (19) and fMRI studies (20-22).” -> p. 5

3. P.4 This is a useful distinction, although there must be discussions and extensions of this approach within the relevant literature since the 1970s? There is a related literature on research around 'concepts as used' (rather than just as linguistic or mental representation) - for example see the edited volume Scientific Concepts and Investigative Practice (Feest & Steinle, 2012).

Thank you for these interesting articles, we added one of the articles from the edited volume “Scientific Concepts and Investigative Practice” to our discussion:

“Consensus on the used operationalisation is not always desirable. For instance, Thomas Kuhn argues that scientific advancement happens through violation of consensus, i.e. a paradigm shift (13)). Moreover, McLeod (14) argues that concepts do not only help to advance scientific knowledge by exactly representing aspects of the world, but through their open-endedness and epistemic vagueness. In his view concepts are not representing theoretical ideas frozen in time but are part of a continual development. In such a way an epistemically vague or fuzzy concept can inspire exact reformulations, as well as construction of experimental techniques for probing and testing it, if it relates to a representation specifying its structure and causal nature. We argue that, in order to advance from fuzzy to exact concept, (13, 14) individual researchers need to notice problematic operationalisations (i.e. there is no overlap between methodology and intended hypothesis) and it should have an effect on the conclusion they personally draw from presented findings. Otherwise, these fuzzy concepts would never be detected, and they could not be advanced over time.-> p.4

4. The idea that there is a law-like relationship between theoretical entities and their observable indicators is at odds with much contemporary philosophy of science literature. One difficulty is that consensus is not always required for successful operationalisation of a concept (see studies of the uses of the concept of the gene, for example). Of course, consensus seems the only way to set up the current experiment.

This is an interesting point. We agree that consensus does not mean that a particular conclusion is correct. However, individual researchers must notice if they see a non-valid operationalization and they should integrate that subjective opinion into the way they evaluate a research claim. To reflect this, we added the following paragraph

We argue that, in order to advance from fuzzy to exact concept, (13, 14) individual researchers need to notice problematic operationalisations (i.e. there is no overlap between methodology and intended hypothesis) and it should have an effect on the conclusion they personally draw from presented findings. Otherwise, these fuzzy concepts would never be detected, and progress could not occur.” -> p.4

5. I have already noted that **the rationale for the choice of research questions should be more explicitly articulated**. At a high level, both hypotheses seem highly plausible. **My concern is, ironically, with how they are operationalised.**

The manipulation of the operationalisation seems fine in each scenario but I still don't quite understand why there are 4 response options. One is obviously the correct one, but how exactly do the other 3 vary? They are described as being 'plausible alternatives' but do they differ in their plausibility in any ways? Are some better fits with the poorer operationalisations? And why 4 (or rather 3, excluding the correct one)?

We constructed these alternatives ourselves with the intent of them being plausible alternatives. Because the relevant comparison is within task between high/medium/low validity conditions, for us it does not matter how well the incorrect alternatives compare to one another (both within and across tasks), only how the response proportions compare across validity conditions.

7. I'm also concerned about a ceiling effect of sorts, with the first task, i.e., that the intended hypotheses will be so easy to infer even when it is plainly clear that the operationalisation is poor. In other words, I'm afraid that the first measure won't discriminate. The quality rating task probably will discriminate, but it seems a rough way to measure understanding of or attention to quality. Giving participants the full set of operationalisations and asking them to rank them might give a fuller picture of their understanding, in a more direct way.

A ceiling effect may occur for the high validity condition, but for the low validity condition participants have no reasonable information to answer correctly, so accuracy should be closer to 25% (guessing) than to 100%. Moreover, a ceiling effect will become likely if the ask participants to state what the intended operationalization was, therefore we will ask participants what they personally would draw conclusion from the presented method and results sections:

"Each research scenario will be presented as published research and the participants will be asked to indicate what they would most likely conclude from the study." -> p. 13

We prefer not to ask for a ranking of the operationalisations, because we are not interested in a rank order for the various interpretations but just in discriminating between picking the correct one and any other. The idea of this paper is 'proof of principle', that is we would like to see whether differences in perceived plausibility of operationalisations do exist.

8. There's one further concern, of a different kind. If the text presented to the participants does not include a definition of the construct the instruments are intended to operationalise (to varying degrees of validity) - in this case, false memory - then it's unclear on whether participants will be using the same concept of false memory when choosing from the options in the multiple choice question. If participants draw on different associations with the concept of false memory (as a type of memory), rather than on false memory as a specified construct (for a given cluster of covarying behaviours reliably attributed to the set of standard indicators), then the differences in how they answer the

multiple choice question may not relate to the operationalisation of the construct so much as the varying uses of a fuzzy (or undefined) concept. I don't know how likely this is, but it does seem like something that could happen.

This problem is inherent in the use of fuzzy concepts, different interpretations of the phenomenon may lead to different answers. To the extent that this would happen, we do not expect it to happen in a systematically different way across validity conditions, and as such should not affect the qualitative pattern of results.

9. Whether the clarity and degree of methodological detail would be sufficient to replicate exactly the proposed experimental procedures and analysis pipeline

This seems sufficiently well covered, but having the exact question wording for task 2 would be preferable.

We agree, and we changed the wording of the two questions now:

“After each presented scenario, we will ask the researcher about the perceived quality of the presented study. Participants will answer the following questions: “ How would you judge the quality of the presented study?” and “ How would you judge the strength of support for the theoretical proposition?” on a scale from 1 (very low) to 10 (very high).” -> **p. 15**

Reviewer 3

I believe that the S1 protocol is adequate in terms of details in the methods and analyses sections, but should be refined to achieve a higher level of reproducibility and prevent undisclosed flexibility that could prove problematic at S2. For example, some major revisions I have recommended include adding more details to the study procedure, correcting a potential error in the group assignment procedure and being explicit about potential data exclusions such as handling missing data. More importantly, since this is a S1 RR protocol I think it is absolutely necessary to have a concise hypotheses paragraph somewhere in the paper than can be linked with respective statistical tests without leaving room for subjective interpretation by the reader.

Introduction section

Line 53, p. 3. Repeated use of ‘often’- in my opinion it would read better if you replace with “social sciences, which primarily depend on the investigation of abstract concepts” and you could give a simple example (one word), “abstract concepts, such as [...]”.

We agree that the sentence could be improved and changed it in our paper.

“Thus, the issue of defining concepts and establishing their relationship to observations is especially relevant in the social sciences, which primarily depend on the investigation of abstract concepts, such as creativity or intelligence.” -> **p. 3**

Line 35, p. 4. Very optional, but it would be good to remind the reader in this last sentence of the paragraph that this can undermine the reproducibility (or replicability) of research findings.

We agree and changed it.

“Instead, a different construct underlies the relation of a dependent (e.g. taste for pop songs) and independent variable (e.g. performance on an IQ test), which disappears when changing one’s measurement and experimental procedure, undermining the reproducibility of a research finding” -> p. 4

Line 47, p. 4. Use of “without” with “no” - replace with “with no..” or only “without”. Sentence may need rephrasing. Do you mean to say that it would not be possible for these theoretical statements to become a system.., without deductive fertility and derivation(s?) chains to observational statements? This is quite specific language and sentence could be clearer for the reader to follow. In my opinion, an example would be great in this paragraph, but again it’s optional - you could extend the creativity example (what is a derivation chain in this context- put in a parenthesis so the word count isn’t affected).

This is a great idea, and we changed text accordingly.

“Without deductive fertility derivation chains to observational statements would not be possible. We would, for example have a hard time deriving a person’s intelligence from their favourite pop song.” -> p.4

Line 28, p. 5. “the relation” - not entirely clear, maybe replace with relationship between [X] and [Y], or this effect.

This has been changed in the text.

“Instead, a different construct underlies the relation of a dependent (e.g. taste for pop songs) and independent variable (e.g. performance on an IQ test), which disappears when changing one’s measurement and experimental procedure, undermining the reproducibility of a research finding.”-> p. 5

Line 46, p. 5. “employed measures” to avoid repetition of “used” (optional).

This has been changed in the text.

“But even when the stimuli are a valid sample from the stimuli population, previous researchers have noted that the employed measures can lack a clear theoretical foundation, thus increasing the resulting flexibility during data analysis (i.e. allowing selective reporting (17)).”-> p. 5

Line 48, p. 5. I would suggest the addition of a citation in the parenthesis where selective reporting is mentioned, just in case the reader is interested to know more about this QRP.

We agree. We added a reference to John et al (2012) as reference [17]:

Line 53, p. 5. have “demonstrated”.. which is a paradigm “that measures” (or supposedly measures if this is part of the argument), allows (researchers) to operationalise “the” severity of..

This has been changed in the paper.

“For example, Elson and colleagues (19) have demonstrated that the Competitive Reaction Time Task, which is a paradigm that measures aggressive behaviour, allows researchers to

operationalise the severity of aggressive behaviour via a noise blast's volume, duration or a composite score of both" -> p. 6

Line 15, p. 6. of "such" or "these" operationalisations

Agreed. We have changed this as follows.

"In the following study, we will not investigate the lack of clear operationalisation in specific studies, but we will empirically test researchers' interpretation of these operationalisations."
-> p. 6

Line 26, p. 6. concept-as-intended and concept-as-determined terms could be mentioned here too (very optional).

We think this would make the sentence unnecessarily complicated, and therefore we decided to leave the sentence as it was.

Line 34, p. 6. "published" research to increase specificity (e.g., see line 42, p. 11)- you can also cut this into one sentence, "we are going to conduct a preregistered study that will gather empirical evidence..."

We agree and changed that.

"To fill this gap, we are going to conduct a preregistered study in which we will investigate to what extent researchers consider the validity of operationalisation when drawing conclusions about empirical findings." -> p. 6

Method section

Line 30, p. 7. Refer reader to section where the group assignment is explained (see X). Also, major point here is that Groups A and B are introduced but reading through the next subsection (e.g., as stated in Fig 1 caption) there appears to be three groups in the design (A, B and C), so the group assignment procedure may be incorrect? Thank you for this helpful remark, yes we changed the description of the assignment procedure.

"These journals were chosen to represent a sample of researchers in diverse fields of psychology (experimental, social, neuro, and clinical), a sampling strategy previously used by Cramer et al. (23). All duplicates will be removed. Participants who do not respond after two weeks will receive a reminder. One third of the participants will be randomly assigned to Group A, one third will be assigned to Group B and the last third will be assigned to group C. We will randomize the email list, so that approximately each group gets assigned the same number of email addresses." -> p. 7

Line 22, p. 8. Italicise "doctor"

This has been changed:

"The results showed that unseen words that were semantically related to the memorized list (i.e., the word 'doctor' which was not presented in a list of medically related words) increases

a false impression of recognition, in comparison to semantically unrelated words. That is, the original study claims that participants falsely remembered 'doctor'.p. 8

Line 17, p. 10. Dependent variable may be clearer.

This has been changed:

"The results indicated that an easier to read font colour increased the perceptual fluency and thus the truthfulness rating of a sentence (i.e. "How true is this sentence?")." -> p. 10

Line 25, p. 15. It would be good to mention what you will tell participants about the aims of the study, as according to instructions or information they may be primed to be more skeptical about the validity of the presented methods in the different scenarios (more relevant for task 2).

Yes, we agree. See changes below.

"The invitation email will state: "Dear (specified name), we are researchers at the university of Groningen (Netherlands) and are interested in investigating how researchers evaluate studies. The following link will advance you to an online experiment, in which you will be presented with three different studies. If you should choose to participate, your task would be to link the presented study to the correct hypothesis and judge the quality of the study. Participation will take approximately 15 minutes. We would appreciate your response to our invitation. With kind regards (name of researchers)." The reminder email will state the same text, starting with: "Dear (specified name), hereby we would like to send you a reminder for our study investigating your perception of three different studies. Please follow this link..."

-> p. 7

Line 8, p. 16. BF explanations are the opposite if you refer to BF 10 - BF of 1/10 refers to support for the null compared to the alternative hypothesis, and similarly for BF > 10, there is support for H1 compared to H0.. unless I missed something here.

Agreed, this has been changed in the text.

"The Bayes factor (BF_{01}) is the relative ratio of the likelihood of the data, given the null hypothesis, and the likelihood of the data, given the alternative hypothesis. In contrast with NHST, the BF_{01} allows to quantify evidence in favour of the null hypothesis (2 9). For instance, a BF_{01} of 10 indicates that the observed data are 10 times more likely under the null hypothesis than under the alternative hypothesis; a BF_{01} of 1 indicates that the observed data are equally likely under both hypotheses (i.e., data does not favour one hypotheses over the other) and a BF_{01} of 1/10 indicates that the observed data are 10 times more likely under the alternative hypothesis than under the null hypothesis. In line with the guidelines of Lee and Wagenmakers (30), we will interpret a BF_{01} between 1-3 as anecdotal evidence, a BF_{01} between 3-10 as moderate evidence, 10-30 as strong evidence, 30-100 as very strong and > 100 as extreme evidence in favour of H0." -> p. 17/18

Line 31, p. 17. Are you planning to do any follow-up analyses after the ANOVA that could be deemed confirmatory? If yes, these should be stated. Accordingly, what will you do if

the ANOVA assumptions are not met? I lack the expertise for the proportions test to comment further on that.

We are not planning to do a post-hoc test. We are only reporting Bayes factors, we are not reporting posterior odds. We changed the text accordingly

“We will report descriptives (means, standard deviations), we will not report posterior odds, and we are not planning to do a post-hoc test. In case the assumptions of the ANOVA are not met, we will instead conduct a Bayesian Kruskal-Wallis test. Moreover, we will conduct an exploratory analysis (Bayesian ANCOVA) including the familiarity of the researcher with the presented scenario.” -> **p. 19**

Line 21, p. 18. The use of ‘conclusive’ here suggests that if you end at max N (6000), evidence will be interpreted as inconclusive even if BF for H1, for example, is 8 or 9 (or 10 - this could be ≥ 10 if this is not true). If this is not the case, it should be explained in the S1 protocol to avoid interpretation issues later on.

Thank you for this comment. We added further information on how BF lower than 10 will be interpreted:

“In line with the guidelines of Lee and Wagenmakers (30), we will interpret a BF_{01} between 1-3 as anecdotal evidence, a BF_{01} between 3-10 as moderate evidence, 10-30 as strong evidence, 30-100 as very strong and > 100 as extreme evidence in favour for H_0 .” -> **p. 18**

Line 30, p. 18. Remove quotation mark.

This has been changed in the paper.

“We will stop collecting data after having gathered 640 participants, or after having targeted 6000 participants, even if the final Bayes factor is still between 1/10 and 10.” -> **p. 20**

Other comments on methods & analyses.

Major issues

1. Missing data, outliers. Although it may not seem fundamental to the S1 protocol, it is important to account for all possibilities when processing and analysing the data. From the methods section I wasn’t sure if all responses will be required (i.e., they cannot proceed if they haven’t answered a question) for participants. Are you going to treat participants who consistently respond in the same way across all scenarios as outliers? Are you planning to clean the data in any way (e.g., incomplete responses)? Even if missing data or outliers are not handled in any way, this needs to be clearly stated in an S1 RR protocol.

The following paragraph has been added to the paper to make this clearer:

“Participants who took less than 2 minutes to fill in the study and who did not finish the study will be excluded. We will make every question mandatory for proceeding. We will not exclude participants who are familiar with the presented studies.” -> **p. 8**

2. Study procedure. From the methods section I was also not entirely confident to reproduce the study are participants answering a survey online? If yes, how is this administered (are you using specific software)? Are the three scenarios presented in

random order across participants or in fixed order (1. Memory, 2. Fluency, 3. Dissonance)? Please add more details if possible about how participants conduct these tasks after they have been reached via email.

“We will send emails with a link to an online questionnaire, made with the online survey platform Qualtrics, to the corresponding authors of all articles published in 2015, 2016, 2017, 2018 and 2019 from the following journals..”

“The invitation email will state: “Dear (specified name), we are researchers at the university of Groningen (Netherlands) and are interested in investigating how researchers evaluate studies. The following link will advance you to an online experiment, in which you will be presented with three different studies. If you should choose to participate, your task would be to link the presented study to the correct hypothesis and judge the quality of the study. Participation will take approximately 15 minutes. We would appreciate your response to our invitation. With kind regards (name of researchers).” The reminder email will state the same text, starting with: “Dear (specified name), hereby we would like to send you a reminder for our study investigating your perception of three different studies. Please follow this link...” -> **p. 7/8**

3. Hypotheses. I understand that you have specified your research questions and planned analyses, but I am missing a hypotheses subsection or related statements. Even if these are indirectly stated in the methods, it is important for RRs to have a specific paragraph with stated confirmatory hypotheses (a priori), which relate to exclusive statistical tests.

Yes, this is a crucial point!

“We hypothesize that researchers are less capable to deduce an original hypothesis from a less valid operationalization (less valid condition leads to less correct deduction of original hypothesis). However, we assume that the less valid operationalisation does not affect the perceived quality of a study. Thus, we hypothesize that less operationalisation validity does not affect the rating of a study’s quality.” -> **p. 7**

Minor issues

4. Other. How do we account for the confound of experience? Authors that are more senior may have knowledge about the observed effects and related theory. I think you may need to ask them directly after completion- were you aware of this task or related theory and research for task 1 (multiple choice- hypotheses). This relates to point 1 (are you going to exclude non-naive participants)?

“After participants completed the study, we will ask them for their familiarity with each presented scenario (0 = not at all, 10 = very familiar).”-> **p. 8**

Participants who took less than 2 minutes to fill in the study and who did not finish the study will be excluded. We will make every question mandatory for proceeding. We will not exclude participants who are familiar with the presented studies.”-> **p.8**

“Moreover, we will conduct an exploratory analysis (Bayesian ANCOVA) including the familiarity of the researcher with the presented scenario.” -> **p. 20**

5. Consistency. The protocol mentions that the study will have two parts and that participants will answer a multiple-choice question and subsequently rate the quality of the study. However, on line 17 p. 17 you mention “specification of H1 for both experiments”, which I found confusing - what are the two experiments - this may suggest that some participants will only do part one and others the second part.

This is a typo. We changed this in the paper.

“For the specification of H1 for each presented scenario...” -> **p. 19**

The participant-related sections (Sampling plan, Participants, Study procedure) are not in close proximity in the paper so I would suggest moving the Sampling plan closer to the Participants paragraph and refer to relevant analyses in the main text (see X for details on Bayesian analyses), but this is entirely optional.

We moved the study procedure section closer to the participant section.

6. Test of three proportions. This section was a bit hard to follow but it is great that you provided all the important details for the test in line with RR guidelines. The only issue I found here relates to my third comment on Hypotheses. You have the description shown below, but as a reader I would like to know how this reflects your hypothesis descriptively - e.g., the proportion of participants who pick the correct hypothesis [...]. Not clearly stating your hypothesis in relation to your planned test leaves room for interpretation which does not ensure the highest level of reproducibility. Thankfully, this is easy to fix by adding a small subsection on specific hypotheses. “Our first analysis will focus on estimation and uses $\sigma = 1$. Our second analysis will focus on testing and quantifies the extent to which the data support the $H_0: \psi = 0$ versus the one-sided $H_1: \psi > 0$. This one-sided alternative indicates a positive value for ψ , resulting in $p_1 > p_2 > p_3$ ”

This has been changed in the text.

“Our first analysis will focus on estimation and uses $\sigma = 1$. Our second analysis will focus on testing and quantifies the extent to which the data support the $H_0: \psi = 0$ versus the one-sided $H_1: \psi > 0$. This one-sided alternative indicates a positive value for ψ , resulting in $p_1 > p_2 > p_3$, thus reflecting our hypothesis that a less valid operationalisation leads to lower rates of deduction of the intended hypothesis. For the specification of H1 for each presented scenario we initially use $\sigma = 0.4$ (i.e., a mildly informative prior; 28), truncated at zero to take into account that the alternative hypothesis is one-sided, which gives $H_1: \psi \sim N(0, 0.4^2)$. We will make use of MCMC to sample from the posterior and will initially use 10,000 iterations. We will visually check convergence for each parameter by using trace plots and check whether the Gelman-Rubin diagnostic value (R-hat; 29) is lower than 1.001. The R code for the analysis can be found online at <https://osf.io/z4qab/>. “ -> **P. 20**.

Appendix D

03.06.2021

Dear Dr. Chambers,

My co-authors and I would like to submit the stage 2 manuscript RSOS-191354.R2 entitled "When numbers fail: Do researchers agree on operationalisation of published research?". On page 27, you can find the URL for the stage 1 manuscript, data, materials, and statistical/qualitative analysis. We hereby confirm that no data for the pre-registered study were collected prior to the date of IPA. Moreover, the completed experiment has been executed and analysed in the manner originally approved, with the following three minor changes.

First, according to the preregistered sampling plan, we had to approach more researchers, because the response rate was lower than expected (p. 6). Secondly, there was an essential typo in the answer option for Task 2: "increase" should have been "decrease" (p. 12). Thirdly, for scenario 3, we noted that the original study used two monotonous tasks (for 30 minutes), instead of one monotonous task (for 60 minutes). We changed the presentation of the scenario accordingly (p. 10/11). None of these changes affected the essence of our design, nor our research questions and hypotheses.

Finally note that contrary to the stage II author instructions, we made a few linguistic edits to the introduction. We emphasise that no substantive changes were made and to facilitate inspection, we have marked all changes compared to the stage 1 manuscript in blue.

We look forward to your comments.

With kind regards,

Matthias Haucke, Rink Hoekstra and Don van Ravenzwaaij

Appendix E

Review for RSOS-191354.R2

This is a very interesting study that has implications for the improvement of psychological science - from training researchers to shifting attention in open research developments (e.g., to focus more on measurement and underlying constructs/theory in psychology). In line with the guidelines for Stage 2 RRs, I have certain observations regarding the completed research and only a few minor comments for revision.

General observations

- The Introduction, rationale and stated hypotheses are the same as the approved Stage 1 protocol, with the exception of an added clarification for the authors' expectations about the study outcomes (page 8, lines 52-60). This clarification does not present a conflicting, or novel, interpretation of the hypotheses.
- The authors have adhered to the Stage 1 protocol but certain details about the methods seem to be missing from the Stage 2 manuscript, as noted in the minor comments for revision.
- The authors have followed the preregistered analysis plan included in the Stage 1 protocol. I have a minor comment regarding the covariance analyses that are presented in the 'Preregistered analyses' section. At Stage 1 these analyses were labelled 'exploratory'. I believe the authors may refer to preregistered non-confirmatory analyses - i.e., these analyses are not linked with any preregistered hypotheses. For clarity it may be better to present all analyses that do not relate to the hypotheses in the 'Exploratory analyses' section.

Minor comments for revision

Participants section, p. 8.

- The authors provide a link to their OSF project which has several files related to the data collection procedure (e.g., invitation email, R code for power analysis etc) and the exact sampling plan that would be followed after IPA. Given that this is an important part of a Registered Report I believe some information should be presented in the manuscript (e.g., what stopping rule was used; if based on BFs then it is not clear whether this rule was followed at Stage 2). This information was originally included in the Stage 1 protocol. This is optional of course, but even if this information is not added in the final manuscript, the sampling plan (<https://osf.io/v5kb7/>) may require updating to note any deviations that occurred at Stage 2 and remove terms that are not present in the final manuscript (e.g., Experiment 1 and Experiment 2).
- There are certain details missing from this section that were included in the Stage 1 protocol, as for example this sentence: "One third of the participants will be randomly assigned to Group A, one third will be assigned to Group B and the last third will be assigned to group C." It is stated that participants were randomly assigned to groups and details can be found in "the Procedure section below" but there is no such information in that section, which has been moved to the end of the Methods. This may simply be an error (e.g., Research Scenarios instead of Procedure).
- Other information that is missing at Stage 2 relates to data exclusions; see text below from Stage 1 protocol. In line with the previous comment, please include these details (or reference for accessing them) in the final manuscript as even minor details can be important for direct replications. Such details do not necessarily need to be in the main text, but data exclusions can also be presented in an Appendix or online Supplementary Info together with the number of participants in each group (e.g., did you achieve the 1/3 of participants in each group 'rule' specified at Stage 1?).

"Participants who took less than 2 minutes to fill in the study and who did not finish the study will be

excluded. We will make every question mandatory for proceeding. We will not exclude participants who are familiar with the presented studies.”

Line 32, p. 9. Change to past tense “will be presented”.

Lines 47-58, p. 22. The values presented here for the calculation of BFincl could be omitted (or included in a JASP output on OSF). By that I mean, presenting only the final BFincl (i.e., instead of “ $((0.997+5.46e-27)/(0.003+1.41e-30)) / ((0.25+0.25)/(0.25+0.25)) = 332.3$ ” write “332.3”. Instead of using scientific notation, the authors could also use log values with a footnote explaining a key benchmark.

Lines 28-46, p. 25. From the discussion alone it is not clear what the preregistered study hypotheses were and what the results were from the corresponding analyses. I understand that this information can be found by reading the relevant sections but as a reader I would prefer to be reminded which outcomes reflect your preregistered hypotheses and respective analyses and which results, if any, were derived from exploratory analyses.

Line 41, p. 26. Change “construct are often embedded” to plural. Just to note, this is a very interesting idea I was not familiar with and a great addition to the discussion.

Lines 48-55, p. 26. I agree with the authors that this is an important issue for the education of psychological researchers, but I am not entirely sure the presented findings for graduate training specifically are enough to justify this claim. There are many ways in which *undergraduate* students for example can be introduced to measurement-related issues in psychology without the need for specialised/full courses. There are two ways to address this comment: add another example / study where this lack of attention is evidenced or change the wording to reflect the state of evidence for this claim (e.g., use “may” instead of “is” on line 48).

Lines 21-39, p. 28. The authors mention that the results are limited by the three chosen scenarios and this issue can also be tied to the findings from the exploratory analyses on familiarity, which were not mentioned in the discussion. If possible, I think the limitations section would benefit from a brief reference to the potential confound/importance of familiarity and how this can be addressed in future work.

Dear Reviewers,

We would like to thank you for your helpful comments. In the following document you can find each of our response to each point that has been raised.

With kind regards,

Matthias Haucke, Rink Hoekstra and Don van Ravenzwaaij

Reviewer 3:

1. The authors have followed the preregistered analysis plan included in the Stage 1 protocol. I have a minor comment regarding the covariance analyses that are presented in the 'Preregistered analyses' section. At Stage 1 these analyses were labelled 'exploratory'. I believe the authors may refer to preregistered nonconfirmatory analyses - i.e., these analyses are not linked with any preregistered hypotheses. For clarity it may be better to present all analyses that do not relate to the hypotheses in the 'Exploratory analyses' section.

Thank you for this comment. We agree and move this section to the exploratory analyses part. See p. 22.

2. The authors provide a link to their OSF project which has several files related to the data collection procedure (e.g., invitation email, R code for power analysis etc) and the exact sampling plan that would be followed after IPA. Given that this is an important part of a Registered Report I believe some information should be presented in the manuscript (e.g., what stopping rule was used; if based on BFs then it is not clear whether this rule was followed at Stage 2). This information was originally included in the Stage 1 protocol. This is optional of course, but even if this information is not added in the final manuscript, the sampling plan (<https://osf.io/v5kb7/>) may require updating to note any deviations that occurred at Stage 2 and remove terms that are not present in the final manuscript (e.g., Experiment 1 and Experiment 2).

Thank you for this comment. Indeed, the initial response rate was 10% instead of the expected 30%. Therefore, we included 3x as many initial participants to reach the targeted sample size. We added this information to the sampling plan (<https://osf.io/wxef3/>)

“Stage 2 addition

Our initial response rate was 10% instead of 30%, therefore we included 2,981 participants to reach the planned initial participant number of approximately 300. In this batch we reached Bayes Factors higher than 10 and lower than 1/10, therefore we stopped sampling.”

3. There are certain details missing from this section that were included in the Stage 1 protocol, as for example this sentence: “One third of the participants will be randomly assigned to Group A, one third will be assigned to Group B and the last third will be assigned to group C.” It is stated that participants were randomly assigned to groups and details can be found in “the Procedure section below” but there is no such information in that section, which has been moved to the end of the Methods. This may simply be an error (e.g., Research Scenarios instead of Procedure).

Thank you for spotting this typo. Indeed, we meant Research Scenario, not Procedure.

“Participants were randomly assigned to each of three groups (groups are described in the Research Scenarios section below” p. 7

4. Other information that is missing at Stage 2 relates to data exclusions; see text below from Stage 1 protocol. In line with the previous comment, please include these details (or reference for accessing them) in the final manuscript as even minor details can be important for direct replications. Such details do not necessarily need to be in the main text, but data exclusions can also be presented in an Appendix or online Supplementary Info together with the number of participants in each group (e.g., did you achieve the 1/3 of participants in each group ‘rule’ specified at Stage 1?). “Participants who took less than 2 minutes to fill in the study and who did not finish the study will be excluded. We will make every question mandatory for proceeding. We will not exclude participants who are familiar with the presented studies.”

Thank you for these comments. We now added this information to the relevant section:

“In total 325 (10,9%) participants started the online survey, no participant took less than 2 minutes and 66 were excluded due to incomplete responses, resulting in a final sample of 259 participants.” (p. 7)

“There were 85 participants in group A, 90 participants in Group B, and 84 participants in group C.” (p.7)

5. Line 32, p. 9. Change to past tense “will be presented”.

“Each scenario was presented with a different validity condition to each participant (see Figure 1).” (p. 7)

6. Lines 47-58, p. 22. The values presented here for the calculation of BFincl could be omitted (or included in a JASP output on OSF). By that I mean, presenting only the final BFincl (i.e., instead of “ $((0.997+5.46e-27)/(0.003+1.41e-30)) / ((0.25+0.25)/(0.25+0.25)) = 332.3$ ” write “332.3”. Instead of using scientific notation, the authors could also use log values with a footnote explaining a key benchmark.

“Finally, using JASP we conducted two Bayesian ANCOVA with the covariate ‘familiarity of with presented scenario’ and perceived quality and well as support for theoretical proposition as outcomes. We report the inclusion Bayes factor (BFincl), which quantifies the change from prior inclusion odds to posterior inclusion odds and can be interpreted as the evidence in the data for including a predictor (i.e., In this case familiarity with the presented scenario)(31). With perceived quality as an outcome, for Validity Condition we found BFincl = 332.3 and for Familiarity BFincl = $1.8e+26$. With perceived support for theoretical proposition as an outcome, for Validity Condition we found BFincl = 499 and for Familiarity BFincl = $4.6e+16$. Therefore, we can conclude that operationalisation validity condition as well as familiarity influenced perceived quality of the study.” (p. 22)

7. Lines 28-46, p. 25. From the discussion alone it is not clear what the preregistered study hypotheses were and what the results were from the corresponding analyses. I understand that this information can be found by reading the relevant sections but as a reader I would prefer to be reminded which outcomes reflect your preregistered hypotheses and respective analyses and which results, if any, were derived from exploratory analyses.

“ A sign of proper operationalisation is a high interpersonal consensus on how the theoretical terms are linked to observations (12), in line with the preregistered hypothesis we found that researchers are better at inferring the underlying research hypothesis from empirical results in more valid operationalisation scenarios. Therefore, we conclude that we have successfully manipulated the validity of the operationalisation. We found mixed evidence for the preregistered hypothesis stating that the validity of the operationalization affects the perceived quality of the study. An exploratory analysis shows that researchers were less capable to deduce what the tested hypotheses were in the medium and high validity conditions, yet this did not influence either their perception of the quality of the study nor the perceived support for the study’s theoretical proposition. Thus, we found some support for the notion that the validity of the operationalisation does not affect researchers’ evaluation of an empirical result.”(p. 23)

8. Line 41, p. 26. Change “construct are often embedded” to plural. Just to note, this is a very interesting idea I was not familiar with and a great addition to the discussion.

“Moreover, scientists might be hesitant to question established psychological constructs, as constructs are often embedded in “generative entrenchment”, meaning that once a concept has been established (e.g., social anxiety disorder) many other concepts (e.g., fear conditioning), theories (e.g., reinforcement learning) or practices (e.g., cognitive behaviour therapy) depend on it (43).” (p. 24)

9. Lines 48-55, p. 26. I agree with the authors that this is an important issue for the education of psychological researchers, but I am not entirely sure the presented findings for graduate training specifically are enough to justify this claim. There are many ways in which undergraduate students for example can be introduced to measurement-related issues in psychology without the need for specialised/full courses. There are two ways to address this comment: add another example / study where this lack of attention is evidenced or change the wording to reflect the state of evidence for this claim (e.g., use “may” instead of “is” on line 48).

“Finally, there might be a lack of attention on operationalisation during the education of psychological researchers. A review of graduate training in psychology has shown that few departments offered a full course on measurement, such as classical test theory (20–24% depending on the topic) (44). “ (p.24)

10. Lines 21-39, p. 28. The authors mention that the results are limited by the three chosen scenarios and this issue can also be tied to the findings from the exploratory analyses on familiarity, which were not mentioned in the discussion. If possible, I think the limitations section would benefit from a brief reference to the potential confound/importance of familiarity and how this can be addressed in future work.

We agree and now added a discussion of this to the limitations:

“Moreover, we found that familiarity impacts the perceived quality of a study, thus another fruitful avenue might be to develop completely unknown scenarios.” (p.26) .

.